# Bilateral regulation of EGFR activity and local PI(4,5)P$_2$ dynamics in mammalian cells observed with superresolution microscopy

**Mitsuhiro Abe[1], Masataka Yanagawa[1,2], Michio Hiroshima[1,3], Toshihide Kobayashi[1,4], Yasushi Sako[1]\***

[1]Cellular Informatics Laboratory, RIKEN Cluster for Pioneering Research, Wako, Japan; [2]Molecular and Cellular Biochemistry, Graduate School of Pharmaceutical Sciences, Tohoku University, Sendai, Japan; [3]Laboratory of Single Molecule Biology, Graduate School of Frontier Biosciences, Osaka University, Osaka, Japan; [4]Laboratoire de Bioimagerie et Pathologies, UMR 7021 CNRS, Université de Strasbourg, Faculté de Pharmacie, Illkirch, France

**Abstract** Anionic lipid molecules, including phosphatidylinositol-4,5-bisphosphate (PI(4,5)P$_2$), are implicated in the regulation of epidermal growth factor receptor (EGFR). However, the role of the spatiotemporal dynamics of PI(4,5)P$_2$ in the regulation of EGFR activity in living cells is not fully understood, as it is difficult to visualize the local lipid domains around EGFR. Here, we visualized both EGFR and PI(4,5)P$_2$ nanodomains in the plasma membrane of HeLa cells using super-resolution single-molecule microscopy. The EGFR and PI(4,5)P$_2$ nanodomains aggregated before stimulation with epidermal growth factor (EGF) through transient visits of EGFR to the PI(4,5)P$_2$ nanodomains. The degree of coaggregation decreased after EGF stimulation and depended on phospholipase Cγ, the EGFR effector hydrolyzing PI(4,5)P$_2$. Artificial reduction in the PI(4,5)P$_2$ content of the plasma membrane reduced both the dimerization and autophosphorylation of EGFR after stimulation with EGF. Inhibition of PI(4,5)P$_2$ hydrolysis after EGF stimulation decreased phosphorylation of EGFR-Thr654. Thus, EGFR kinase activity and the density of PI(4,5)P$_2$ around EGFR molecules were found to be mutually regulated.

## Editor's evaluation

This paper reports a fundamental study explaining how signalling through a cell-surface receptor can lead to different cellular outputs through changing interactions with membrane lipids. The evidence supporting the conclusions is compelling, with state-of-the-art microscopy, including multicolour super-resolution microscopy and single-molecule imaging, coupled with carefully controlled experiments. This work will be of broad interest to cell biologists, particularly those with interests in cell signalling and lipid biology, in addition to researchers using imaging to study cellular phenomena.

## Introduction

Epidermal growth factor receptor (EGFR) is a receptor tyrosine kinase responsible for cell proliferation and differentiation (*Nyati et al., 2006*; *Olayioye et al., 2000*). EGFR is subdivided into five regions: a large extracellular region, a single-spanning transmembrane I region, an intracellular juxtamembrane (JM) region, a tyrosine kinase region, and a C-terminal tail region (*Lemmon et al., 2014*).

**\*For correspondence:**
sako@riken.jp

**Competing interest:** The authors declare that no competing interests exist.

**eLife digest** Residing on the surface of cells are proteins called receptors, which bind to external molecules. Once activated, receptors undergo various changes that allow them to relay the signal to other components inside the cell that can alter the cell's behavior.

One such protein is the epidermal growth factor receptor (or EGFR for short), which helps regulate cell division and development. When molecules bind to an EGFR, this causes the receptor to attach to another EGFR in the membrane to form a dimer. This dimerization is crucial as it allows the two receptors to add chemicals known as phosphates to each other, which recruit additional proteins that relay the activation signal to downstream targets inside the cell.

Studies have shown that a lipid which sits within the cell membrane, called $PI(4,5)P_2$, helps stabilize the EGFR dimer and aid its activation. However, it is not fully understood exactly how $PI(4,5)P_2$ achieves this.

To investigate, Abe et al. used a super-resolution microscope that can visualize single molecules to examine how $PI(4,5)P_2$ lipids are distributed around the receptor. This revealed that EGFR and $PI(4,5)P_2$ overlap one another to form structures termed 'nanodomains' before the receptor is stimulated. Further experiments showed that the nanodomains promote dimerization and activation of EGFRs. They also provide a surface for downstream molecules to dock on to, making it easier for them to relay signals into the cell.

Abe et al. found that once an EGFR has been stimulated, $PI(4,5)P_2$ is broken down by downstream molecules. This results in fewer nanodomains and induces a process that deactivates the signaling pathway.

The findings of Abe et al. suggest that $PI(4,5)P_2$ enhances EGFR signaling by forming nanodomains which are then dissolved once the receptor has been activated. This aligns with previous studies showing lipids in the cell membrane influence the behavior of receptors similar to EGFRs.

The gene for EGFR, and the receptor itself, have both been shown to display abnormal activity in various human cancers. In the future, the work of Abe et al. may provide new insights into how nanodomains influence this irregular signaling, potentially aiding researchers in discovering new cancer treatments.

After EGFR binds to epidermal growth factor (EGF), the extracellular region adopts conformations favoring the dimerization of the transmembrane helices near their N terminus and the dimerization of the JM region (*Arkhipov et al., 2013*; *Ogiso et al., 2002*), inducing the formation of asymmetric (active) dimers of the intracellular kinase region (*Red Brewer et al., 2009*; *Thiel and Carpenter, 2007*; *Zhang et al., 2006*). This dimerization results in the phosphorylation of several tyrosine residues in the tail region and the recruitment of intracellular signal proteins, such as growth factor receptor-bound protein 2 (GRB2) and phospholipase Cγ (PLCγ) containing Src homology 2 (SH2) and/or phosphotyrosine-binding regions (*Wagner et al., 2013*). The JM region of EGFR plays a crucial role in the conformation-dependent coupling of EGFR's binding to EGF and its activation and dimerization (*Endres et al., 2013*; *Jura et al., 2009b*). The JM region comprises a JM-A (N-terminal half) region, which can form an antiparallel helix dimer, and a JM-B (C-terminal half) region, which makes intramolecular contact with the kinase region (*Jura et al., 2009a*). Both these JM regions contribute to the stable formation of an asymmetric kinase dimer, which is important for kinase activation.

Anionic lipid molecules in the inner leaflet of the plasma membrane are implicated in the dimerization of the JM regions (*Hedger et al., 2015*; *Matsushita et al., 2013*; *McLaughlin et al., 2005*). Multiscale molecular dynamic simulations have suggested that phosphatidylinositol-4,5-bisphosphate ($PI(4,5)P_2$) interacts specifically with the basic residues in the JM-A region, stabilizing the JM-A helices in an orientation away from the membrane surface by binding to $PI(4,5)P_2$ (*Abd Halim et al., 2015*; *Matsushita et al., 2013*). In addition to these in silico analyses, our recent in vitro study using nanodisc techniques suggested that phosphatidylserine (PS) and $PI(4,5)P_2$ stabilize the conformation of the JM-A dimer (*Maeda et al., 2018*; *Maeda et al., 2022*). Apart from these in silico and in vitro studies, the localization and roles of PS and $PI(4,5)P_2$ during EGFR activation have not been fully investigated experimentally in living cells due to the difficulty of visualizing the local lipid domains around EGFR in the plasma membrane.

The main difficulty is that the proposed size of the lipid domain is below the diffraction limit of light. Therefore, conventional fluorescence microscopy cannot image the structure in detail. Super-resolution imaging techniques that break the diffraction limit have become a powerful tool for visualizing cellular structures with unprecedented resolution (*Pujals et al., 2019*). Among these techniques, single-molecule localization microscopy (SMLM), which includes photoactivated localization microscopy, fluorescence photoactivation localization microscopy, stochastic optical reconstruction microscopy (STORM), ground-state depletion microscopy followed by individual molecular return, and direct STORM, allows the construction of super resolved images with high-precision localization of individual fluorophores (*Rust et al., 2006*).

Here, we visualized both the EGFR and PI(4,5)P$_2$ nanodomains in the plasma membrane with SMLM. We found that the PI(4,5)P$_2$ nanodomains and EGFR nanoclusters aggregate together before EGF stimulation. After stimulation, the degree of coaggregation decreases, which depends on PLCγ but not on phosphoinositide 3-kinase (PI3K). The local PI(4,5)P$_2$ around EGFR stabilizes EGFR dimers to increase EGFR autophosphorylation upon stimulation with EGF. The subsequent hydrolysis of PI(4,5)P$_2$ by PLCγ plays a crucial role in the deactivation of EGFR. Our results suggest that autogenous remodeling of the lipid environments regulates EGFR activity.

## Results

### EGF stimulation reduces PI(4,5)P$_2$ nanodomains

To visualize the nanodomains of EGFR and the lipids in the plasma membrane simultaneously, we constructed a three-color SMLM analysis workflow (see Methods, *Figure 1—figure supplement 1*). EGFR was fused with rsKame, a slow-switching Dronpa variant (*Rosenbloom et al., 2014*), at the C-terminus. To exclude the effect of endogenous EGFR, we knocked out endogenous EGFR in HeLa cells, transfected them with EGFR–rsKame, and selected a cell line stably expressing the EGFR–rsKame fusion protein. When we stimulated the selected cell line with EGF, the EGFR–rsKame and extracellular signal-regulated kinase (ERK) in the cell line were phosphorylated to the same levels as in the parental HeLa cells (*Figure 1—figure supplement 2A and B*), indicating that the EGFR–rsKame fusion protein functioned normally. To observe PI(4,5)P$_2$ with SMLM, the PI(4,5)P$_2$-binding peptide PLCδ–PH was fused with a photoactivatable protein, PAmCherry1 (*Abe et al., 2012*), and was transiently expressed in the cell line. To visualize PS with SMLM, the PS-specific peptide evectin-2 (evt–2)–PH (*Uchida et al., 2011*), tagged with HaloTag, was transiently expressed in the cell line, and was labeled with a spontaneous blinking dye, HMSiR (*Takakura et al., 2017*).

After the cells were fixed, sequential three-color SMLM was performed by imaging EGFR–rsKame illuminated with a 488 nm laser; PAmCherry1–PLCδ–PH (hereafter 'PAmCherry–PI(4,5)P$_2$') activated with a 405 nm laser and illuminated with a 532 nm laser; and HMSiR–evt–2–PH (hereafter 'HMSiR–PS') illuminated with a 637 nm laser (*Figure 1—figure supplement 1A*). A single-molecule tracking analysis, drift correction, alignment of the color channels, image reconstitution, a multidistance spatial cluster analysis, and a G-function spatial map (G-SMAP) analysis were then performed automatically according to the workflow shown in the Methods section.

EGFR–rsKame, PAmCherry–PI(4,5)P$_2$, and HMSiR–PS were distributed nonrandomly in the plasma membrane (*Figure 1A*). The G-SMAP analysis revealed that EGFR, PI(4,5)P$_2$, and PS formed nanodomains with diameters ranging from <100 nm to several hundred nanometers. We measured the densities and areas of the nanodomains (*Figure 1B*). Before EGF stimulation, their size was widely distributed, with a median of 0.0052±0.0002 μm$^2$ (~0.04 μm radius, n=8 cells) for the EGFR–rsKame nanodomains. Neither the density nor the area of the EGFR–rsKame nanodomains was affected by EGF stimulation (median of 0.0053±0.0001 μm$^2$ in area, ~0.04 radius, n=10 cells). However, the density of the PAmCherry–PI(4,5)P$_2$ nanodomains decreased after EGF stimulation; the median area of PAmCherry–PI(4,5)P$_2$ increased from 0.0049±0.0001 μm$^2$ (~0.04 μm radius, n=8 cells) to 0.0081±0.0004 μm$^2$ (~0.05 μm radius, n=10 cells, p<0.001) after EGF stimulation. In particular, the size of smaller PAmCherry–PI(4,5)P$_2$ clusters was reduced after EGF stimulation.

We also analyzed the lateral univariate aggregation of the EGFR–rsKame, PAmCherry–PI(4,5)P$_2$, and HMSiR–PS molecules, calculating Ripley's univariate H-function, the derivative of Ripley's univariate K-function (*Figure 1C*) that is frequently used in cluster analyses. The extent of aggregation, the H(r) value, was plotted as a function of the distance between molecules, r, in micrometers. The distance, r,

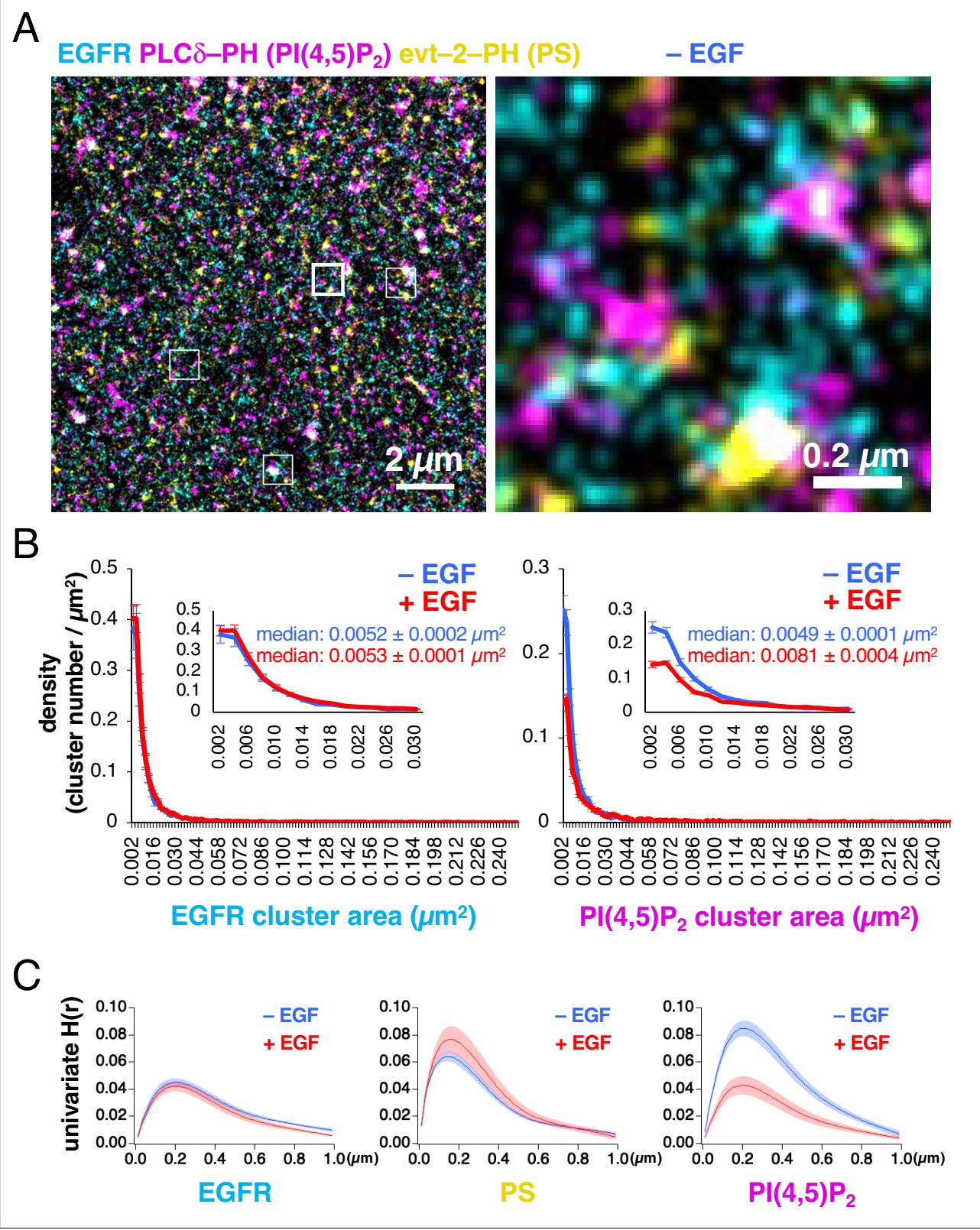

**Figure 1.** Epidermal growth factor receptor (EGFR), PI(4,5)P$_2$, and phosphatidylserine (PS) are distributed nonrandomly in the plasma membrane. (**A**) Images of EGFR–rsKame (cyan), PAmCherry–PI(4,5)P$_2$ (magenta), and HMSiR–PS (yellow) before epidermal growth factor (EGF) stimulation. PAmCherry–PI(4,5)P$_2$ and Halo–evt2–PH (Halo–PS) were transiently expressed in a cell line stably expressing EGFR–rsKame. Cells were incubated in serum-free medium in the presence of the HMSiR–Halo ligand overnight and treated with paraformaldehyde and glutaraldehyde. Left, a typical

*Figure 1 continued on next page*

*Figure 1 continued*

image of 15×15 μm. Right, enlarged image of the square region surrounded by a bold line in the left image. Enlarged images of other square regions surrounded by thin lines in the left image are shown in *Figure 1—figure supplement 2C*. (**B**) Distribution of the densities of EGFR (left) and PI(4,5)P$_2$ cluster areas (right). Cells were incubated in serum-free medium overnight, stimulated with or without 20 nM EGF for 1 min, and treated with paraformaldehyde and glutaraldehyde. From the single-molecule localization microscopy (SMLM) images, the cluster areas and the number of clusters were measured. After the cluster number was normalized to the cell area (density), the cluster density was plotted as a function of the cluster area. Inset, enlarged graphs for the cluster areas of <0.03 μm$^2$. Blue and red indicate before and after EGF stimulation, respectively. Data are means ± SEM of at least eight cells. (**C**) Univariate H(R) values of EGFR–rsKame (left), PAmCherry–PI(4,5)P$_2$ (middle), and HMSiR–PS (right) in cells incubated in the absence (blue) or presence (red) of 20 nM EGF for 1 min. Ripley's univariate H-function was calculated from the SMLM images. Data are means ± SEM of nine (EGFR–rsKame and PAmCherry–PI(4,5)P$_2$) or 10 (HMSiR–PS) cells.

The online version of this article includes the following source data and figure supplement(s) for figure 1:

**Source data 1.** Original image files displayed in *Figure 1A*.

**Source data 2.** Raw data files displayed in *Figure 1B and C*.

**Figure supplement 1.** Multicolor single-molecule localization microscopy (SMLM) analysis workflow.

**Figure supplement 2.** Construction and evaluation of multicolor single-molecule localization microscopy (SMLM).

**Figure supplement 2—source data 1.** PDF file for western blotting analysis displayed in *Figure 1—figure supplement 2A and B*.

**Figure supplement 2—source data 2.** Original files for western blotting analysis displayed in *Figure 1—figure supplement 2A and B*.

that maximizes the H(r) value (referred to as the R-value here) reflects the radius of the area in which the local density of molecules exceeds the random distribution by the largest amount (*Kiskowski et al., 2009*). The R values of EGFR–rsKame, PAmCherry–PI(4,5)P$_2$, and HMSiR–PS were 0.20±0.01 μm (n=9 cells), 0.21±0.01 μm (n=9 cells), and 0.14±0.01 μm (n=10 cells), respectively (*Figure 1C*). Therefore, the EGFR, PI(4,5)P$_2$, and PS molecules accumulated within membrane domains of similar sizes. The aggregations detected in the H-function analysis (~0.2 μm; *Figure 1C*) were larger than those of the nanodomains estimated with the G-function-based clustering analysis of localization (~0.04 μm; *Figure 1B*), indicating a hierarchical domain structure, that is, the formation of nanodomain clusters in the plasma membrane. The areas of aggregation (domain clusters) were not altered by EGF stimulation for 1 min. The R values for EGFR–rsKame, PAmCherry–PI(4,5)P$_2$, and HMSiR–PS were 0.20±0.01 μm (n=9 cells), 0.20±0.01 μm (n=9 cells), and 0.15±0.01 μm (n=9 cells), respectively. The peak H(r) value (referred to as the H(R) value here) indicates the degree of aggregation (*Kiskowski et al., 2009*). The H(R) value of EGFR–rsKame and HMSiR–PS was not significantly different before and after the addition of EGF (*Figure 1C*, left and middle). However, after EGF stimulation, the H(R) value of PAmCherry–PI(4,5)P$_2$ decreased from 0.085±0.006–0.043±0.007 (n=9 cells, p<0.001; *Figure 1C*, right).

## Lateral coaggregation of EGFR and PI(4,5)P$_2$ decreases after EGF stimulation

EGFR and the lipid nanodomains partly colocalized in the plasma membrane (*Figure 1A*, *Figure 1—figure supplement 2C*). To estimate the degree of lateral coaggregation of the two different species of molecules, we calculated Ripley's bivariate H-function (*Liu et al., 2023*; *Zhou et al., 2017*). H(r) was plotted as a function of the distance r (*Figure 2A*), between the two different species of molecules. The peak value of bivariate H(r) (referred to as the bivariate H(R) value here) for EGFR–rsKame and HMSiR–PS was not significantly altered by EGF stimulation (*Figure 2A*, left). The bivariate H(R) value of PAmCherry–PI(4,5)P$_2$ and HMSiR–PS also did not change significantly (*Figure 2A*, middle). However, after EGF stimulation, the bivariate H(R) values of EGFR–rsKame and PAmCherry–PI(4,5)P$_2$ decreased from 0.0126±0.0004–0.0066±0.0005 (n=8 cells, p<0.001). Therefore, EGF stimulation reduced the coaggregation rate of this pair (*Figure 2A*, right).

To confirm these results, we performed the following three experiments. First, we replaced PAmCherry–PI(4,5)P$_2$ with HMSiR–PI(4,5)P$_2$ and observed EGFR–rsKame and HMSiR–PI(4,5)P$_2$ with our SMLM system. EGF stimulation for 2 min significantly reduced the bivariate H(R) value of EGFR–rsKame and HMSiR–PI(4,5)P$_2$ from 0.0133±0.0010 (n=17 cells) to 0.0027±0.0003 (n=20 cells, p<0.001; *Figure 2B*), similar to the results shown in *Figure 2A*, right. Even 0.5 min stimulation with EGF induced a decrease in the bivariate H(R) value of EGFR–rsKame and HMSiR–PI(4,5)P$_2$. Second, we replaced PLCδ–PH with TubbyC, another biosensor for PI(4,5)P$_2$, because the PI(4,5)P$_2$ biosensors

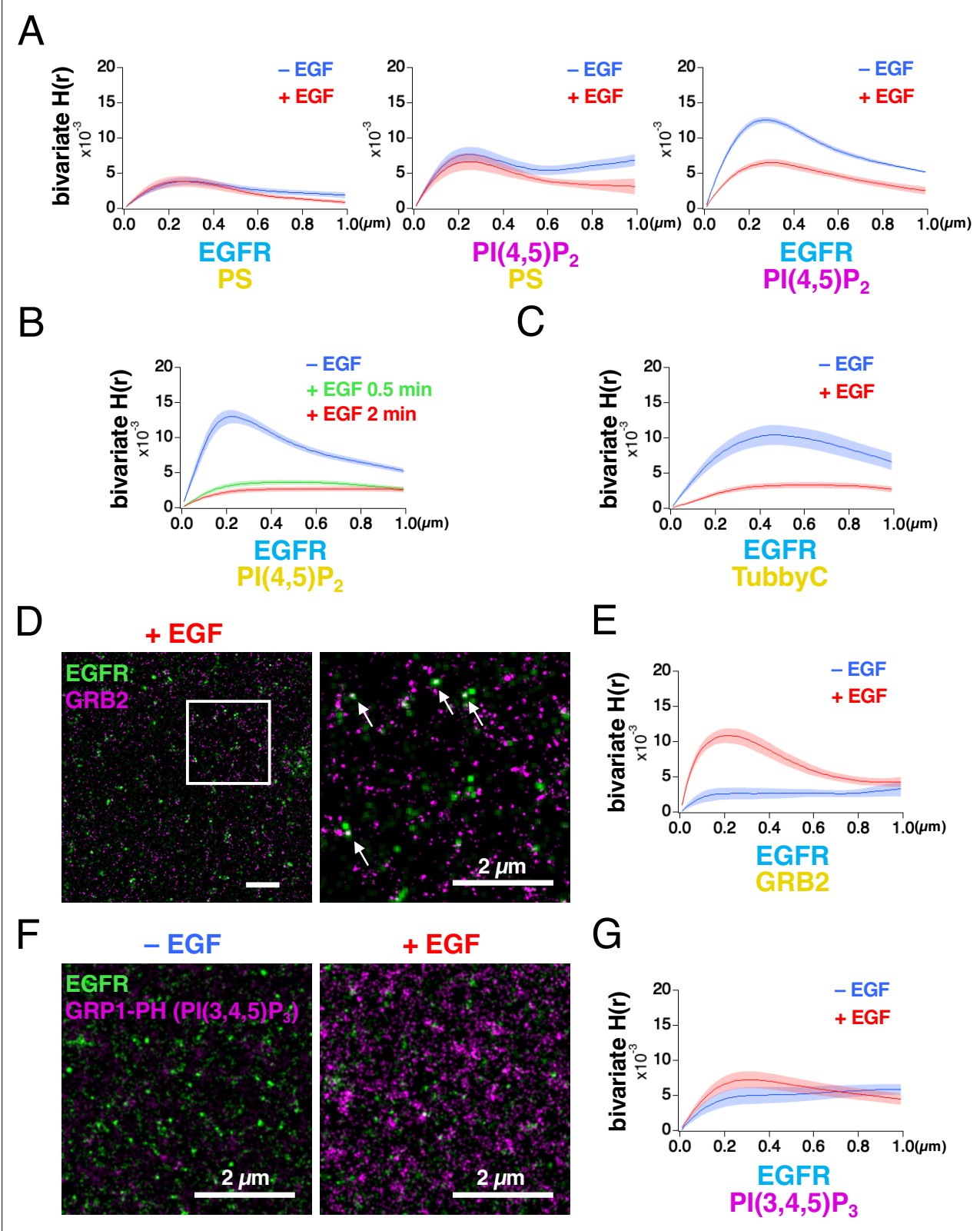

**Figure 2.** Bivariate H(r) calculated from SMLM images reveals that lateral coaggregation of epidermal growth factor receptor (EGFR) and PI(4,5)P$_2$ decreases after epidermal growth factor (EGF) stimulation. (**A**) Bivariate H(r) value of EGFR–rsKame and HMSiR–PS, PAmCherry–PI(4,5)P$_2$ and HMSiR–PS, and EGFR–rsKame and PAmCherry–PI(4,5)P$_2$ before (blue) and after incubation with 20 nM EGF for 1 min (red). Cells were prepared as in *Figure 1*. Ripley's bivariate H-function was calculated from the single-molecule localization microscopy (SMLM) images. Data are means ± SEM of eight cells.

*Figure 2 continued on next page*

*Figure 2 continued*

(**B**) Bivariate H(r) value of EGFR–rsKame and HMSiR–PI(4,5)P$_2$ before (blue) and after incubation with 20 nM EGF for 0.5 min (green) or 2 min (red). Halo–PI(4,5)P$_2$ was transiently expressed in a cell line stably expressing EGFR–rsKame. Cells were incubated in serum-free medium in the presence of the HMSiR–Halo ligand overnight, stimulated with or without 20 nM EGF for the indicated times, and treated with paraformaldehyde and glutaraldehyde. Data are means ± SEM of 17 (without EGF stimulation), 10 (with EGF stimulation for 0.5 min), or 20 cells (with EGF stimulation for 2 min). (**C**) Bivariate H(r) value of EGFR–rsKame and HMSiR–TubbyC before (blue) and after incubation with 20 nM EGF for 2 min (red). Halo–TubbyC was transiently expressed in a cell line stably expressing EGFR–rsKame. Data are means ± SEM of 8 (without EGF stimulation) or 13 cells (with EGF stimulation for 2 min). (**D**) Images of EGFR–rsKame (green) and HMSiR–GRB2 (magenta) after EGF stimulation for 2 min (right). Halo–GRB2 was transiently expressed in a cell line stably expressing EGFR–rsKame. Right, enlarged image of the square region surrounded by a bold line in the left image. Arrows indicate EGFR and GRB2 overlaps in the images. (**E**) Bivariate H(r) value of EGFR–rsKame and HMSiR–GRB2 before (blue) and after incubation with 20 nM EGF for 2 min (red). Halo–GRB2 was transiently expressed in a cell line stably expressing EGFR–rsKame. Data are means ± SEM of 10 cells. (**F**) Images of EGFR–rsKame (green) and PAmCherry–PI(3,4,5)P$_3$ (magenta) before (left) and after EGF stimulation for 2 min (right). PAmCherry1–GRP1-PH (PAmCherry–PI(3,4,5)P$_3$) was transiently expressed in a cell line stably expressing EGFR–rsKame. (**G**) Bivariate H(r) value of EGFR–rsKame, and PAmCherry–PI(3,4,5)P$_3$ before (blue) and after incubation with 20 nM EGF for 2 min (red). Data are means ± SEM of 10 (before EGF stimulation) or nine cells (after 20 nM EGF stimulation).

The online version of this article includes the following source data for figure 2:

**Source data 1.** Raw data files displayed in *Figure 2A, B, C, E and G*.

**Source data 2.** Original image files displayed in *Figure 2D and F*.

show differences in mobility in the plasma membrane (*Pacheco et al., 2023*). Similar to the results in *Figure 2B*, EGF stimulation significantly reduced the bivariate H(R) value of EGFR–rsKame and HMSiR–TubbyC from 0.0108±0.0013 (n=8 cells) to 0.0038±0.0004 (n=13 cells, p<0.001; *Figure 2C*). Third, we observed EGFR–rsKame and HMSiR–GRB2 with our SMLM system and measured the bivariate H(R) value. We found several EGFR and GRB2 overlaps in the SMLM images after EGF stimulation for 2 min (*Figure 2D*). EGF stimulation increased the bivariate H(R) value of EGFR–rsKame and HMSiR–GRB2 from 0.0027±0.0008 (n=10 cells) to 0.0110±0.0010 (n=10 cells, p<0.001; *Figure 2E*). These values are consistent with the finding that cytoplasmic GRB2 is recruited to phosphorylated EGFR after EGF stimulation (*Yoshizawa et al., 2021*). These three experiments support the idea that the H-function analysis of the SMLM data reflects the degree of coaggregation, and that EGF stimulation reduced the colocalization rate of EGFR and PI(4,5)P$_2$.

Because PI(4,5)P$_2$ and PI(3,4,5)P$_3$ have similar characteristics but play different roles in the plasma membrane (*Insall and Weiner, 2001*), we also examined the coaggregation of EGFR and PI(3,4,5)P$_3$ before and after EGF stimulation. PAmCherry1–GRP1-PH (hereafter 'PAmCherry–PI(3,4,5)P$_3$'), which specifically interacts with PI(3,4,5)P$_3$, was expressed in the EGFR–rsKame stable cell line (*Figure 2F*) and the bivariate H(r) values of EGFR–rsKame and PamCherry–PI(3,4,5)P$_3$ were measured (*Figure 2G*). The density of PAmCherry–PI(3,4,5)P$_3$ was small with a low bivariate H(R) value before EGF stimulation. After EGF stimulation, the bivariate H(R) value was still low (*Figure 2G*, red), although, consistent with a previous report (*Malek et al., 2017*), the number of PI(3,4,5)P$_3$ nanodomains stained with PAmCherry–PI(3,4,5)P$_3$ increased (*Figure 2F*). Thus, the nanodomains of PI(4,5)P$_2$ and PI(3,4,5)P$_3$ displayed distinct characteristics in their interactions with EGFR before and after EGF stimulation.

## Partial and transient overlap between EGFR and PI(4,5)P$_2$ decreases after EGF stimulation in living cells

Because the cells used in SMLM were artificially fixed with paraformaldehyde and glutaraldehyde, we examined the colocalization of EGFR and PI(4,5)P$_2$ in living cells using single-molecule tracking (SMT) analysis (*Figure 3*, *Figure 3—figure supplement 1*). For SMT analysis, we tagged the cytoplasmic tail of EGFR with an enhanced green fluorescent protein (EGFR–GFP) (*Hiroshima et al., 2018*; *Yasui et al., 2018*) and expressed it in EGFR-knockout (KO) HeLa cells at low levels. EGFR–GFP in the cells prepared for SMT analysis was phosphorylated to the same level as in the parental HeLa cells after EGF stimulation (*Figure 3—figure supplement 1A*), suggesting that EGFR–GFP functioned normally. After the EGFR–GFP and Halo–PI(4,5)P$_2$ fusion proteins were expressed in EGFR-KO HeLa cells, Halo–PI(4,5)P$_2$ was labeled with Janelia Fluor 549 (JF549). These proteins were observed with dual-color total internal reflection fluorescence for SMT analysis (*Figure 3A*). The movements of the fluorescent particles were classified into three motional modes (immobile, slow-mobile, and fast-mobile) according to the Variational Bayesian-Hidden Markov Model (*Yanagawa and Sako, 2021*; *Yasui et al., 2018*). The proportion of EGFR in the immobile fraction increased and the ratio in the fast-mobile

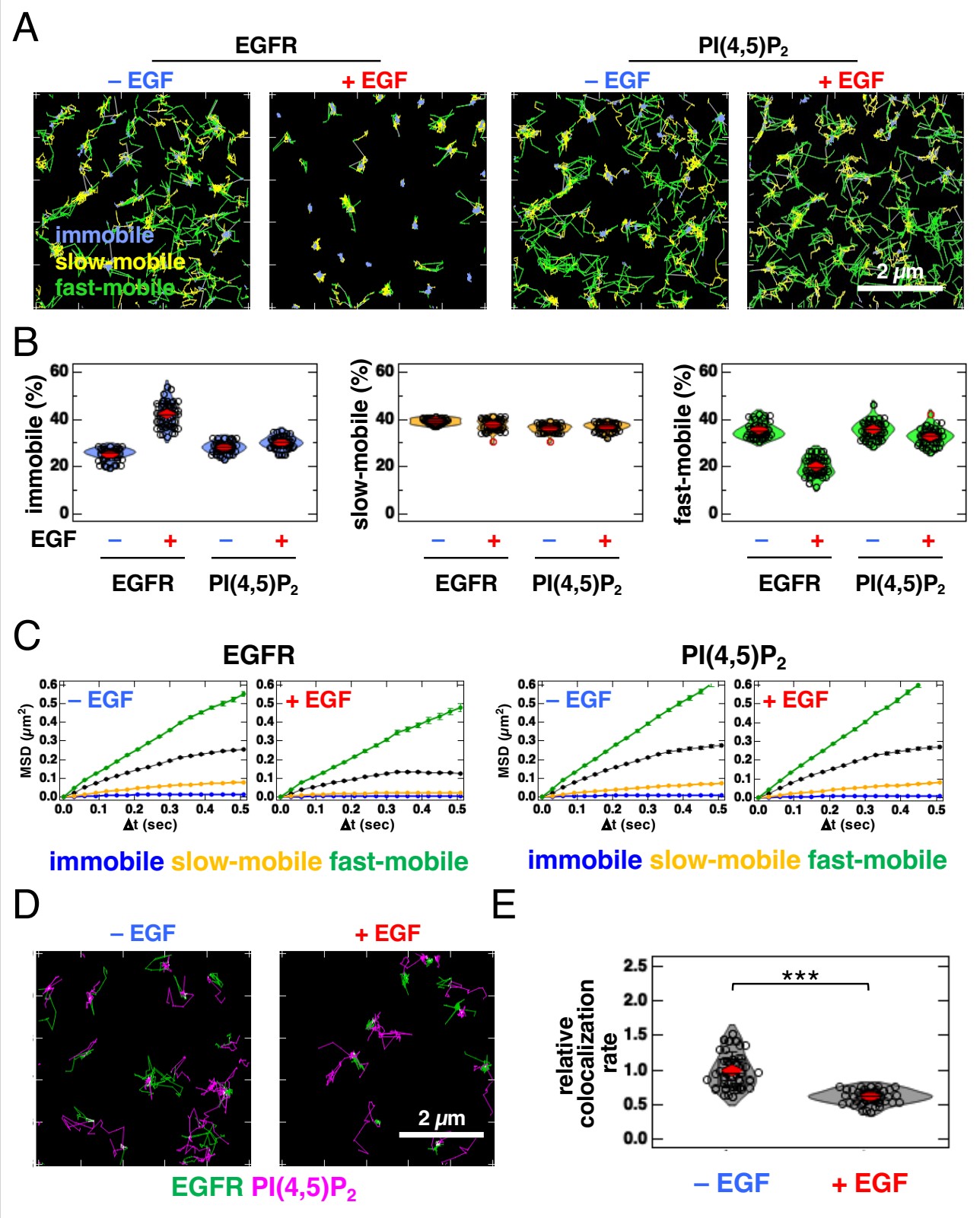

**Figure 3.** Single-molecule tracking (SMT) analysis reveals that partial and transient overlap between epidermal growth factor receptor (EGFR) and PI(4,5)P₂ decreases after epidermal growth factor (EGF) stimulation. (**A**) Trajectories of EGFR–GFP (left) and JF549–PI(4,5)P₂ (right) over 6 s. EGFR–GFP and Halo–PLCδ–PH (Halo–PI(4,5)P₂) were transiently expressed in EGFR-KO HeLa cells. After the cells were incubated in a serum-free medium overnight, Halo–PI(4,5)P₂ was stained with JF549–Halo ligand. EGFR–GFP and JF549–PI(4,5)P₂ were observed in 30 cells at a time resolution of 30 ms for 6 s

*Figure 3 continued on next page*

*Figure 3 continued*

before EGF stimulation. After stimulation with 20 nM EGF, EGFR–GFP and JF549–PI(4,5)P$_2$ were observed in the same cells between 1 and 5 min after the addition of EGF. SMT analysis was fractionated into the immobile (blue), slow-mobile (yellow), and fast-mobile (green) fractions. (**B**) Proportions of immobile (blue), slow-mobile (yellow), and fast-mobile (green) fractions. (**C**) Mean square displacement per unit time (MSD-Δt) plots of the trajectories of EGFR–EGFP and JF549–PI(4,5)P$_2$. (**D**) Trajectories of EGFR–GFP (green), JF549–PI(4,5)P$_2$ (magenta), and colocalization (white) during 6 s before (left) and after incubation with 20 nM EGF (right) in the same cell. (**E**) Relative colocalization rates of EGFR–GFP and JF549–PI(4,5)P$_2$ before and after incubation with 20 nM EGF. To consider the differences in expression levels among cells, the colocalization rate was divided by the densities of EGFR–GFP and JF549–PI(4,5)P$_2$ and normalized to the mean value obtained before EGF stimulation. Violin plots show the mean value and distribution of 30 cells. \*\*\*p<0.001 on Welch's t-test.

The online version of this article includes the following source data and figure supplement(s) for figure 3:

**Figure supplement 1.** Single-molecule tracking (SMT) analysis of epidermal growth factor receptor (EGFR)–EGFP and JF549–PI(4,5)P$_2$.

**Figure supplement 1—source data 1.** PDF file for western blotting analysis displayed in *Figure 3—figure supplement 1A*.

**Figure supplement 1—source data 2.** Original files for western blotting analysis displayed in *Figure 3—figure supplement 1A*.

**Figure supplement 2.** Time-dependent colocalization of epidermal growth factor receptor (EGFR)-PI(4,5)P$_2$ and EGFR-growth factor receptor-bound protein 2 (GRB2).

fraction decreased after EGF stimulation (*Figure 3B*) at the same level as that reported previously for EGFR–GFP (*Hiroshima et al., 2018*; *Yasui et al., 2018*). The proportion of PI(4,5)P$_2$ in each fraction did not change significantly after EGF stimulation. The mean square displacement per unit time (MSD-Δt) of the lateral movements indicated that the diffusion coefficient of EGFR decreased after EGF stimulation, whereas that of PI(4,5)P$_2$ remained unchanged (*Figure 3C*, *Figure 3—figure supplement 1B*). All diffusion modes displayed subdiffusion. The confinement length of EGFR was also reduced after EGF stimulation, whereas that of PI(4,5)P$_2$ was not (*Figure 3—figure supplement 1C*).

Next, we examined the colocalization of EGFR and PI(4,5)P$_2$. We found partial and transient overlaps in the trajectory images of EGFR and PI(4,5)P$_2$, before and after EGF stimulation (*Figure 3D*; *Video 1*). To consider the differences in expression levels among cells, we normalized the original colocalization rate (*Yanagawa and Sako, 2021*) to the densities of EGFR and PI(4,5)P$_2$ (i.e. particle number/cell area). The normalized data show that the colocalization of EGFR and PI(4,5)P$_2$ decreased after EGF stimulation (*Figure 3E*). Among the three motional modes, the colocalization rate in the slow-mobile and fast-mobile fractions decreased after EGF stimulation, whereas the rate in the immobile fraction did not change significantly (*Figure 3—figure supplement 1D*). We also examined the colocalization of EGFR and TubbyC (*Figure 3—figure supplement 1E*). The colocalization rates of EGFR and TubbyC were similar to those of EGFR and PLCδ–PH before EGF stimulation and decreased to rates comparable to those of EGFR and PLCδ–PH after EGF stimulation (*Figure 3—figure supplement 1F*). The colocalization rate of EGFR and PI(4,5)P$_2$ decreased significantly by 0.5 min after EGF stimulation and remained low for at least 5 min (*Figure 3—figure supplement 2A*). Under the same experimental conditions, the colocalization rate of EGFR and GRB2 increased by 0.5 min after EGF stimulation and remained high for at least 5 min (*Figure 3—figure supplement 2B*). These results suggest that the binding of EGF to EGFR and the subsequent reaction occurred within 0.5 min under our experimental conditions. The transient colocalization of EGFR and PI(4,5)P$_2$ observed in SMT analysis was consistent with the small spatial overlap of the nanodomains of these two molecular species in the SMLM images.

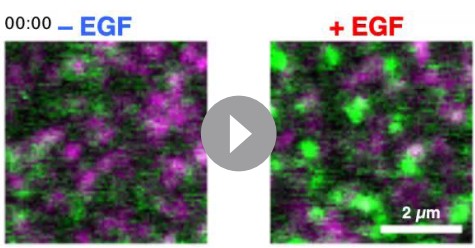

**Video 1.** Lateral colocalization rate of epidermal growth factor receptor (EGFR) and PI(4,5)P$_2$ decreases after epidermal growth factor (EGF) stimulation. EGFR–GFP (green) and Halo–PI(4,5)P$_2$ (magenta) were transiently expressed in EGFR-KO HeLa cells. After the cells were incubated in a serum-free medium overnight, Halo–PI(4,5)P$_2$ was stained with JF549–Halo ligand. EGFR–GFP and JF549–PI(4,5)P$_2$ were observed at a time resolution of 30 ms for 6 s before EGF stimulation (left). After stimulation with 20 nM EGF, EGFR–GFP and JF549–PI(4,5)P$_2$ were observed in the same cell 2 min after the addition of EGF (right).
https://elifesciences.org/articles/101652/figures#video1

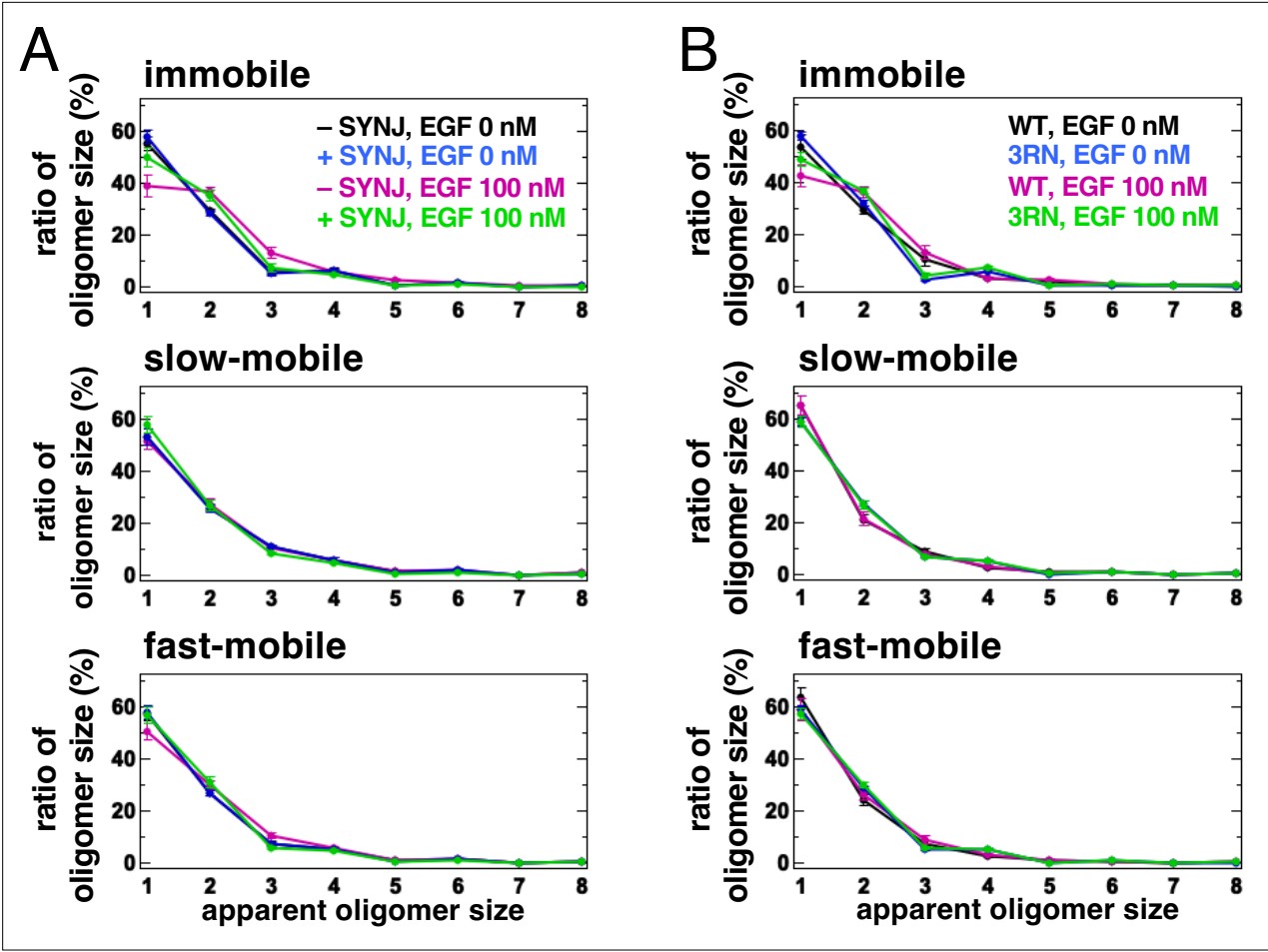

**Figure 4.** Single-molecule tracking (SMT) analysis shows that PI(4,5)P$_2$ is important for stabilizing epidermal growth factor receptor (EGFR) dimers. (**A**) SMT analysis of EGFR–SF650 in control and synaptojanin (SYNJ)-expressing cells. Only EGFR–Halo or both EGFR-Halo and green fluorescent protein (GFP)–SYNJ were transiently expressed in Chinese Hamster Ovary (CHO)-K1 cells. After cells were incubated in a serum-free medium overnight, EGFR–Halo was stained with the SF650–Halo ligand. EGFR–SF650 was observed in control (n=40) and SYNJ-expressing cells (n=40) at a time resolution of 30 ms before epidermal growth factor (EGF) stimulation. Following stimulation with 100 nM EGF, EGFR–SF650 was observed in the same cells between 1 and 6 min after the addition of EGF. (**B**) SMT analysis of EGFR(WT)–SF650 and EGFR(3RN)–SF650. EGFR–Halo or EGFR(3RN)–Halo was transiently expressed in CHO-K1 cells. After the cells were incubated in a serum-free medium overnight, EGFR–Halo was stained with the SF650–Halo ligand. EGFR(WT)–SF650 (n=40) and EGFR(3RN)–SF650 (n=45) were observed as described in (**A**). Data are means ± SEM.

The online version of this article includes the following source data and figure supplement(s) for figure 4:

**Figure supplement 1.** Single-molecule tracking (SMT) analysis of epidermal growth factor receptor (EGFR) under PI(4,5)P$_2$-depleted condition.

**Figure supplement 2.** Single-molecule tracking (SMT) analysis of epidermal growth factor receptor (EGFR) under PI(4,5)P$_2$-interaction-depleted condition.

**Figure supplement 2—source data 1.** Original image files are displayed in *Figure 4—figure supplement 2C*.

## PI(4,5)P$_2$ is important for stabilizing the EGFR dimer

In silico and in vitro analyses have suggested that PI(4,5)P$_2$ is involved in the dimerization of EGFR (*Abd Halim et al., 2015*; *Maeda et al., 2018*; *Maeda et al., 2022*; *Matsushita et al., 2013*). The aggregation of PI(4,5)P$_2$ with EGFR in the plasma membrane may help to stabilize the dimers of EGFR after its association with EGF. To test this, we reduced the content of PI(4,5)P$_2$ by expressing synaptojanin (SYNJ), a phosphatase that hydrolyzes PI(4,5)P$_2$, and measured the size of the EGFR–Halo oligomer in EGFR-null Chinese Hamster Ovary (CHO)-K1 and EGFR-KO HeLa cells by SMT analysis (*Figure 4A*, *Figure 4—figure supplement 1A*). The cytoplasmic expression of SYNJ, decreases the amount of PI(4,5)P$_2$ in the plasma membrane (*Field et al., 2005*). EGFR–Halo was labeled with Sara-Fluor 650T (SF650). In the immobile fraction of control CHO-K1 cells, 55.1±2.6% (n=40 cells) of the

EGFR particles were monomers, whereas 29.7±0.8% (n=40 cells) were preformed dimers before EGF stimulation (*Figure 4A*, *Figure 4—figure supplement 1B and C*). After EGF stimulation, the monomers decreased to 38.8±4.1% (n=40 cells) and the dimers increased to 36.9±1.8% (n=40 cells) in the control cells. The slow- and fast-mobile fractions showed little change in EGFR oligomer size. However, in the immobile fraction of SYNJ-expressing cells, the monomers decreased slightly even after EGF stimulation (57.6±2.7% before and 50.3±3.4% after EGF stimulation, n=40 cells). Similar suppression of EGFR dimers by SYNJ expression was observed in the immobile fraction of EGFR-KO HeLa cells (*Figure 4—figure supplement 1A*). The mean density and mean intensity of EGFR–SF650 in the SYNJ-expressing cells before EGF stimulation were similar to those in the control cells (*Figure 4—figure supplement 1D and E*), indicating that the expression levels of EGFR–SF650 were not significantly altered in SYNJ-expressing cells. These results suggest that $PI(4,5)P_2$ is important for stabilizing EGFR dimers after EGF stimulation.

Molecular dynamics simulations have shown that the conversion of three arginine residues to asparagine residues in the EGFR JM region (EGFR(3RN)) abolishes the interaction between EGFR and $PI(4,5)P_2$ (*Abd Halim et al., 2015*). Therefore, we examined the oligomer size of EGFR(3RN) in CHO-K1 cells before and after EGF stimulation (*Figure 4B*, *Figure 4—figure supplement 2A and B*). Similar to the oligomer size of EGFR(WT), that of EGFR(3RN) showed little change in the slow- and fast-mobile fractions. The monomer in the immobile fraction constituted 48.9±2.5% of EGFR(3RN) (n=45 cells) after EGF stimulation (*Figure 4B*), which was similar to that in the SYNJ-expressing cells (*Figure 4A*). The relative intensity of Cy5–EGF bound to EGFR did not differ significantly between EGFR(WT) and EGFR(3RN), suggesting that the 3RN mutation does not affect the binding of EGFR to EGF (*Figure 4—figure supplement 2C and D*). These results suggest that the interaction of EGFR with $PI(4,5)P_2$ is important for the dimerization of EGFR.

In SMT, molecular-level interactions of dimers cannot be detectable. Therefore, we confirmed these direct interactions with a biochemical analysis using a crosslinker (*Figure 5A and B*). Cells were treated with 0.2 or 20 nM EGF, which is below or above the dissociation constant of 2–6 nM (*Sugiyama et al., 2023*), respectively, and cell surface EGFR was crosslinked with EGFR using bis(sulfosuccinimidyl)suberate. The dimer fraction of EGFR was small before EGF stimulation but increased after stimulation with 0.2 or 20 nM EGF. Under both conditions, the expression of SYNJ reduced the amount of EGFR dimer (*Figure 5A and B*), suggesting that $PI(4,5)P_2$ is involved in stabilizing the dimerization of EGFR.

Because the expression of SYNJ does not completely abolish $PI(4,5)P_2$ in the cell (*Field et al., 2005*), we could not determine whether $PI(4,5)P_2$ is essential for the dimerization of EGFR. To test this, we examined whether EGFR(3RN) forms dimers after EGF stimulation (*Figure 5C and D*). After stimulation with 0.2 or 20 nM EGF, the amount of EGFR(3RN) dimers increased. If the interaction with $PI(4,5)P_2$ was not completely abolished in EGFR(3RN)-expressing cells, the amount of EGFR(3RN) dimers should decrease when $PI(4,5)P_2$ was reduced. There was no additive effect on the dimerization of EGFR(3RN) when SYNJ was expressed (*Figure 5C and D*), suggesting that the interaction of EGFR(3RN) with $PI(4,5)P_2$ was rigorously blocked. Therefore, $PI(4,5)P_2$ is not essential for, but positively regulates EGFR dimerization.

The dimerization of EGFR leads to the phosphorylation of tyrosine residues in its tail region (*Wagner et al., 2013*). We examined whether the reduced dimerization of EGFR under $PI(4,5)P_2$-interaction-depleted conditions resulted in defective phosphorylation of EGFR (*Figure 5E and F*). The measurement of Tyr1068-phosphorylated EGFR after stimulation with 20 nM EGF indicated that the phosphorylation of EGFR decreased when SYNJ was overexpressed in cells. We also found that the Tyr1068 phosphorylation in EGFR(3RN) decreased relative to that in EGFR(WT) after stimulation with 20 nM EGF. However, after simulation with 0.2 nM EGF, no significant difference in the phosphorylation of EGFR was observed between the cells as EGFR was less phosphorylated under the current experimental conditions. These results confirm that $PI(4,5)P_2$ is important for not only the dimerization of EGFR but also its subsequent phosphorylation.

## Extent of reduced coaggregation depends on PLCγ but not on PI3K

A biochemical analysis showed that the amount of $PI(4,5)P_2$ is transiently reduced after EGF stimulation in cells (*Malek et al., 2017*). Two kinds of enzymes are activated by EGFR and reduce the amount of $PI(4,5)P_2$ after EGF stimulation (*Wells, 1999*). First, PI3K phosphorylates $PI(4,5)P_2$ to produce

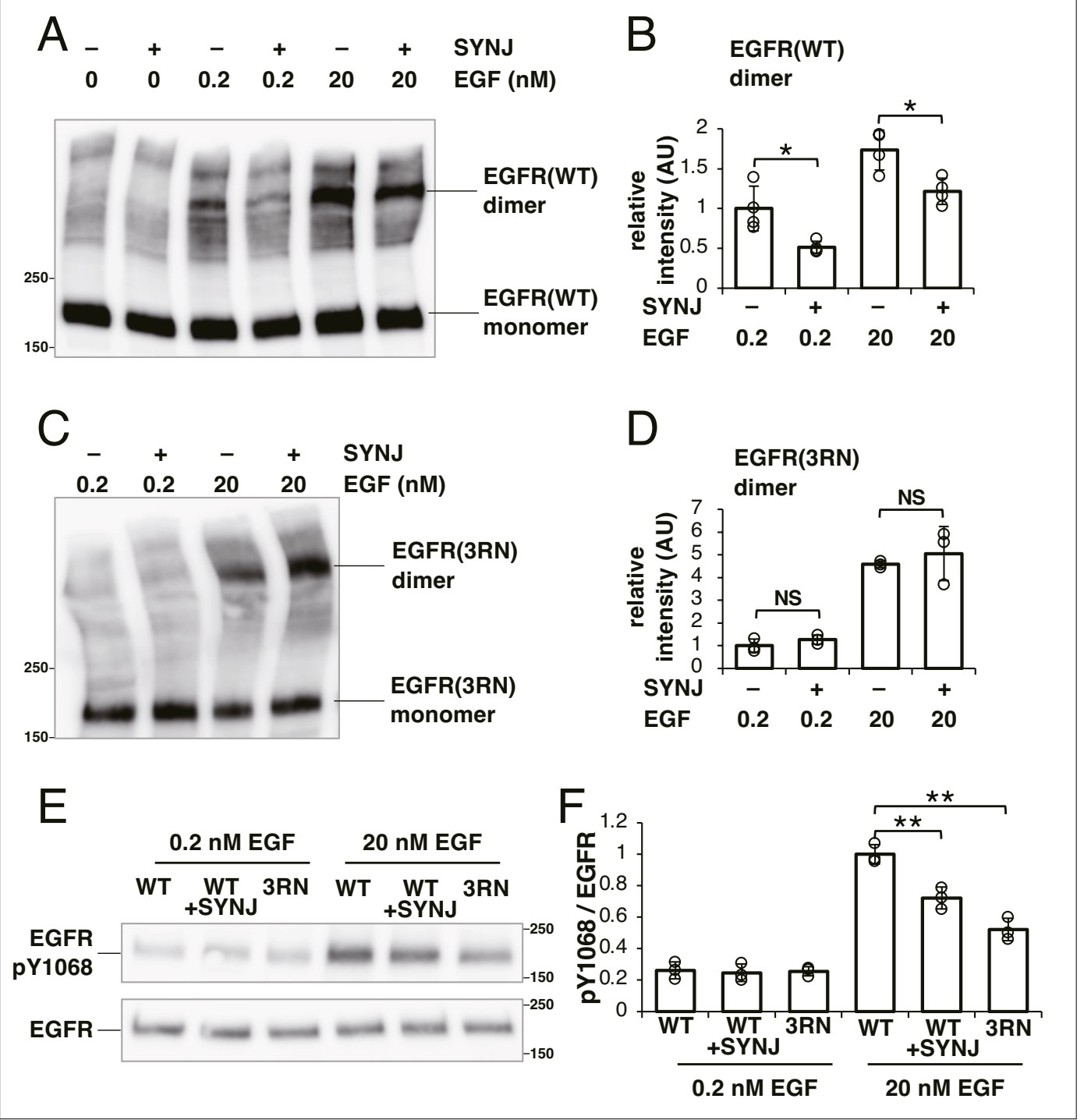

**Figure 5.** PI(4,5)P$_2$ is important for not only the dimerization of epidermal growth factor receptor (EGFR) but also its subsequent phosphorylation.
(**A**) Western blotting analysis of crosslinked EGFR. After EGFR-knockout (KO) cells were transfected with only EGFR–Halo or both EGFR–Halo and green fluorescent protein (GFP)–synaptojanin (SYNJ), the cells were incubated in a serum-free medium overnight. The cells were then stimulated with 0, 0.2, or 20 nM epidermal growth factor (EGF), and treated with a crosslinker for 1 hr. EGFR–Halo was detected with an anti-EGFR antibody. (**B**) Amounts of EGFR–Halo dimer. Relative intensity was normalized to the mean intensity of lane #3 in (**A**). Data are means ± SD of four experiments. (**C**) Western blotting analysis of crosslinked EGFR(3RN). After EGFR-KO cells were transfected with only EGFR(3RN)–Halo or both EGFR(3RN)–Halo and GFP–SYNJ, the cells were incubated in a serum-free medium overnight. The cells were then stimulated with 0.2 or 20 nM EGF, and treated with a crosslinker for 1 hr. EGFR(3RN)–Halo was detected with an anti-EGFR antibody. (**D**) Amounts of EGFR(3RN)–Halo dimer. Relative intensity was normalized to the mean intensity of lane #1 in (**C**). Data are means ± SD of three experiments. (**E**) Western blotting analysis of phosphorylated EGFR and total EGFR. After EGFR-KO cells were transfected with EGFR(WT)–Halo, EGFR(WT)–Halo, and GFP–SYNJ, or EGFR(3RN)–Halo, the cells were incubated in serum-free

*Figure 5 continued on next page*

*Figure 5 continued*

medium overnight and stimulated with 0.2 nM or 20 nM EGF for 2 min. Phospho-EGFR and total EGFR were detected with anti-pY1068 EGFR and EGFR, respectively. (**F**) Ratio of phosphorylated-Tyr1068 EGFR/total EGFR. The ratio was normalized to the mean value of lane #4 in (**E**). Data are means ± SD of three experiments. *p<0.05, **p<0.01, NS (not significant) on Welch's t-test.

The online version of this article includes the following source data for figure 5:

**Source data 1.** PDF file for western blotting analysis are displayed in *Figure 5A, C and E*.

**Source data 2.** Original files for western blotting analysis are displayed in *Figure 5A, C and E*.

PI(3,4,5)P$_3$ through the heterodimerization of ERBB3 with EGFR (*Hellyer et al., 1998*). Second, phosphoinositide-specific PLCγ hydrolyzes PI(4,5)P$_2$ to produce diacylglycerol (DAG) and inositol trisphosphate (IP$_3$) through its direct physical association with EGFR via the SH2 regions of PLCγ (*Carpenter and Ji, 1999*). We investigated the protein responsible for the dissociation of EGFR from the PI(4,5)P$_2$ nanodomains after EGF stimulation. We inhibited either PI3K or PLCγ and measured the bivariate H(r) value of EGFR and PI(4,5)P$_2$ after EGF stimulation.

The addition of wortmannin, a PI3K inhibitor (*Yano et al., 1993*), diminished the PI3K-dependent phosphorylation of AKT after EGF stimulation (*Figure 6A*). The expression of a dominant negative fragment of PLCγ containing the SH2, SH3, and PLC-inhibitory regions (referred to as 'DN PLCγ' here) inhibits PLCγ activity (*Banan et al., 2001*; *Homma and Takenawa, 1992*). We found that the expression of DN PLCγ reduced DAG production after EGF stimulation compared with that in the non-DN-PLCγ-expressing control cells, indicating that the expressed fragment inhibited PLCγ activity (*Figure 6B*).

Under these conditions, we observed EGFR and PI(4,5)P$_2$ in the plasma membrane with SMLM after EGF stimulation. To coexpress the PI(4,5)P$_2$ probe and DN PLCγ in the same cells, we used a plasmid containing an internal ribosome entry site (IRES). Measurement of Ripley's bivariate H-function revealed that no significant difference in the bivariate H(R) value of EGFR and PI(4,5)P$_2$ was observed between the wortmannin-treated (0.0102±0.0010, n=7 cells) and -untreated cells (0.0080±0.0006, n=7 cells), suggesting that PI3K is not important for the disaggregation of EGFR and PI(4,5)P$_2$ (*Figure 6C*). However, the high bivariate H(R) value of EGFR and PI(4,5)P$_2$ was maintained in the DN-PLCγ-expressing cells (0.0225±0.0015, n=8 cells, p<0.001) after EGF stimulation, suggesting that PLCγ is responsible for the reduced coaggregation of EGFR and PI(4,5)P$_2$ (*Figure 6C*).

We examined whether EGFR activation is truly associated with its coaggregation with PLCγ by observing phosphorylated EGFR and PI(4,5)P$_2$ in the plasma membranes of wortmannin-treated or DN-PLCγ-expressing cells. After phosphorylated EGFR was immunostained with anti-EGFR (pY1068) and HMSiR-labeled antibodies in PAmCherry–PI(4,5)P$_2$-expressing cells after EGF stimulation, the bivariate H(r) value of phospho-EGFR and PI(4,5)P$_2$ was measured (*Figure 6D*). The bivariate H(R) values of phospho-EGFR and PI(4,5)P$_2$ in the control, wortmannin-treated, and DN-PLCγ-expressing cells were 0.0080±0.0006 (n=7 cells), 0.0093±0.0010 (n=7 cells), and 0.0193±0.0012 (n=8 cells), respectively. In the SMLM images of DN-PLCγ-expressing cells, overlapping of the nanodomains of phospho-EGFR and PI(4,5)P$_2$ was observed in the submicrometer coaggregation domains indicated in the bivariate H-function analysis (*Figure 6E*).

We found that the univariate H(R) values of PI(4,5)P$_2$ were larger in the wortmannin-treated cells (0.071±0.003, n=9 cells) and DN-PLCγ-expressing cells (0.056±0.005, n=9 cells) than in the control cells (0.024±0.002, n=9; *Figure 6—figure supplement 1A*). These results were expected from the inhibition of the metabolism of PI(4,5)P$_2$ in the wortmannin-treated and DN-PLCγ-expressing cells (*Figure 6A and B*). Interestingly, the R-value was smaller in the wortmannin-treated cells (0.131±0.013 µm, n=9 cells) than in the DN-PLCγ-expressing cells (0.252±0.005 µm, n=9 cells, *Figure 6—figure supplement 1A*), suggesting that PI3K and PLCγ affect the PI(4,5)P$_2$ nanodomains in the plasma membrane differently. We detected no significant difference in the univariate H(R) values of pY1068 between the cells (*Figure 6—figure supplement 1B*). These results support the conclusion that of the two enzymes PI3K and PLCγ, PLCγ mainly affects the reduction in the degree of coaggregation of EGFR and PI(4,5)P$_2$ after EGFR phosphorylation.

ERBB3 directly activates PI3K through canonical p85 binding motifs in ERBB3 (*Soltoff et al., 1994*). One possible explanation for the different effects of these two enzymes is that PLCγ hydrolyzes PI(4,5)P$_2$ around EGFR, whereas PI3K phosphorylates PI(4,5)P$_2$ around the heterodimer of ERBB3 and EGFR

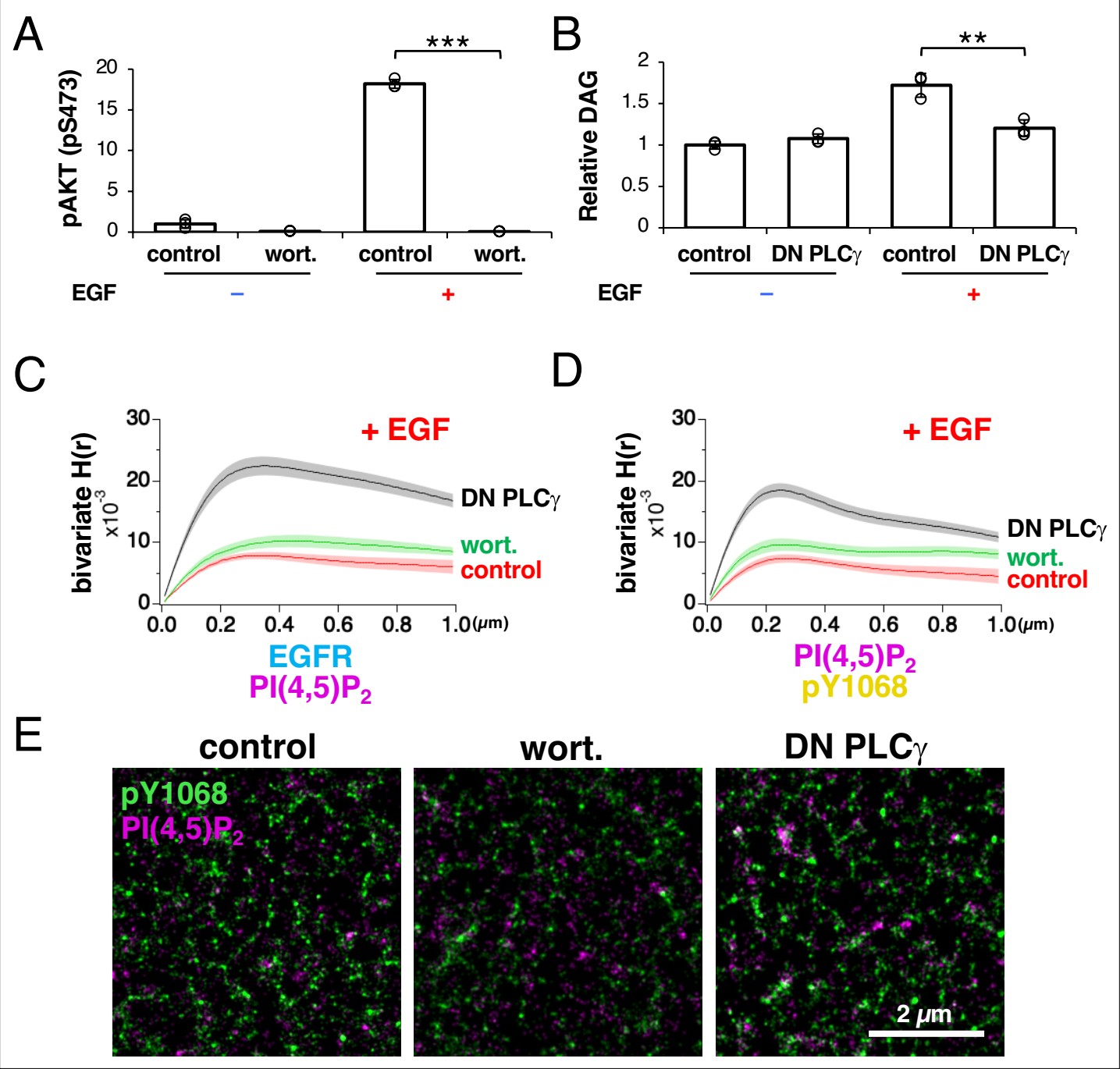

**Figure 6.** Phospholipase Cγ (PLCγ) is responsible for epidermal growth factor receptor (EGFR) dissociation from PI(4,5)P$_2$ nanodomains after EGFR is phosphorylated. (**A**) Effect of wortmannin on phosphorylation of AKT. A cell line stably expressing EGFR–rsKame was incubated in a serum-free medium overnight and treated with 10 μM wortmannin for 1 hr. The cells were then stimulated with or without 20 nM EGF for 1 min. Phosphorylated AKT was detected with an anti-phospho-AKT (Ser473) antibody in a western blotting analysis. Relative intensity was normalized to the mean value of the control cells without EGF stimulation. Data are means ± SD of three experiments. (**B**) Effect of DN PLCγ on diacylglycerol (DAG) production. DN PLCγ was transiently expressed in a cell line stably expressing EGFR–rsKame. Cells were incubated in a serum-free medium overnight and treated with or without 20 nM EGF for 1 min. Amounts of DAG were measured with a DAG assay kit. Relative intensity was normalized to the mean intensity of the control cells without EGF stimulation. Data are means ± SD of three experiments. (**C**) Bivariate H(r) value of EGFR–rsKame and PAmCherry–PI(4,5)P$_2$ in control (red), wortmannin-treated (green), and DN-PLCγ-expressing cells (black) after incubation with 20 nM EGF for 1 min. In control and wortmannin-treated cells, PAmCherry1–PLCδ–PH was transiently expressed in a cell line stably expressing EGFR–rsKame. To co-express PAmCherry1–PLCδ–PH and DN PLCγ in the same cells, plasmid DEST40/PAmCherry1–PLCδ–PH–IRES–DN PLCγ was transfected to the cell line stably expressing EGFR–rsKame. Cells were incubated in a serum-free medium overnight and treated with (control and DN PLCγ) or without 10 μM wortmannin for 1 h (wortmannin). The cells

*Figure 6 continued on next page*

*Figure 6 continued*

were then stimulated with 20 nM EGF for 1 min and treated with paraformaldehyde and glutaraldehyde. Data are means ± SEM of seven (control and wortmannin) or eight cells (DN PLCγ). (**D**) Bivariate H(r) value of PAmCherry–PI(4,5)P$_2$ and HMSiR–pY1068 in control (red), wortmannin-treated (green), and DN-PLCγ-expressing cells (black) after incubation with 20 nM EGF for 1 min. After cells were prepared as described in (**E**), phosphorylated EGFR was immunostained with anti-EGFR (pY1068) and HMSiR-labeled antibodies. Data are means ± SEM of six (control), seven (wortmannin), or eight cells (DN PLCγ). (**E**) Images of PAmCherry–PI(4,5)P$_2$ (magenta) and HMSiR-labeled pY1068 EGFR (green) after stimulation with EGF for 1 min. Cells were prepared as described in (**F**). **p<0.01, ***p<0.001 on Welch's t-test.

The online version of this article includes the following source data and figure supplement(s) for figure 6:

**Source data 1.** Raw data files are displayed in *Figure 6C and D*.

**Source data 2.** Original image files are displayed in *Figure 6E*.

**Figure supplement 1.** Phosphoinositide 3-kinase (PI3K) is not responsible for epidermal growth factor receptor (EGFR) dissociation from PI(4,5)P$_2$ nanodomains after EGFR is phosphorylated.

**Figure supplement 1—source data 1.** Raw data files are displayed in *Figure 6—figure supplement 1A and B*.

**Figure supplement 1—source data 2.** PDF file for western blotting analysis are displayed in *Figure 6—figure supplement 1C*.

**Figure supplement 1—source data 3.** Original files for western blotting analysis displayed in *Figure 6—figure supplement 1C*.

in the plasma membrane. In EGFR KO and ERBB3 KO cells, little AKT phosphorylation was detected after EGF stimulation (*Figure 6—figure supplement 1C and D*). EGFR–Halo expression restored AKT phosphorylation after EGF stimulation in EGFR KO cells, but not in ERBB3 KO cells. Therefore, both ERBB3 and EGFR were required for EGF-induced AKT phosphorylation under our experimental conditions. SMT analysis indicated that the colocalization rate of PLCγ–PI3K was lower than that of EGFR–PLCγ or EGFR–PI3K (*Figure 6—figure supplement 1E and F*). These results support the idea that PLCγ and PI3K react differently to PI(4,5)P$_2$ nanodomains in the plasma membrane.

## Hydrolysis of PI(4,5)P$_2$ by PLCγ is involved in the deactivation of EGFR

As shown in *Figure 6*, we found that PLCγ is responsible for the reduced coaggregation of EGFR and PI(4,5)P$_2$ after EGF stimulation. PLCγ produces DAG, which is known to activate protein kinase C (PKC) (*Steinberg, 2008*). PKC activation is known to induce the phosphorylation of EGFR-Thr654 (*Hunter et al., 1984*), which is involved in the deactivation of EGFR. Therefore, we speculated that the hydrolysis of PI(4,5)P$_2$ around EGFR molecules by PLCγ upon EGF stimulation has a role in the deactivation of EGFR via PKC. To test this hypothesis, we measured the amount of Thr654-phosphorylated EGFR under PLCγ-inhibited conditions (*Figure 7A and B*). We found that EGFR Thr654 was less phosphorylated in DN-PLCγ-expressing cells than in the control cells after EGF stimulation. The amount of Thr654-phosphorylated EGFR was also reduced in PLCG1-knockdown (KD) cells. The reduced phosphorylation of EGFR-Thr654 in PLCG1-KD cells was restored by adding 4β-phorbol 12-myristate 13-acetate (PMA), which is a DAG mimetic (*Figure 7C and D*), suggesting that the phosphorylation of Thr654 depends on DAG produced by PLCG1. Moreover, when the tyrosine kinase activity of EGFR was inhibited by AG1478, the phosphorylation of Thr654 was blocked in the control cells as well (*Figure 7C and D*), which suggests that the phosphorylation of EGFR-Thr654 requires autophosphorylation of EGFR.

To exclude the possibility that the inhibition of PLCγ reduces the autophosphorylation of EGFR, we measured the phosphorylation of EGFR-Tyr1068 in DN-PLCγ-expressing and PLCG1-KD cells (*Figure 7A and B*). Consistent with previous literature on mouse fibroblasts (*Ji et al., 1998*), the levels of Tyr1068-phosphorylated EGFR were not significantly altered in the DN-PLCγ-expressing or PLCG1-KD cells, suggesting that the inhibition of PLCγ does not block the kinase activity of EGFR. These results suggest that the autophosphorylation of EGFR activates PLCγ after EGF stimulation, inducing the phosphorylation of EGFR-Thr654 in a DAG-dependent manner.

## Discussion

We examined the dynamics and functional roles of PI(4,5)P$_2$ as the boundary and/or neighboring lipid of EGFR in the activation of EGFR by EGF stimulation. Superresolution microscopy revealed that EGFR and PI(4,5)P$_2$ form nanodomains that partially overlap before EGF stimulation. The nanodomains also form domain clusters and the extent of coaggregation of EGFR and PI(4,5)P$_2$ molecules in a

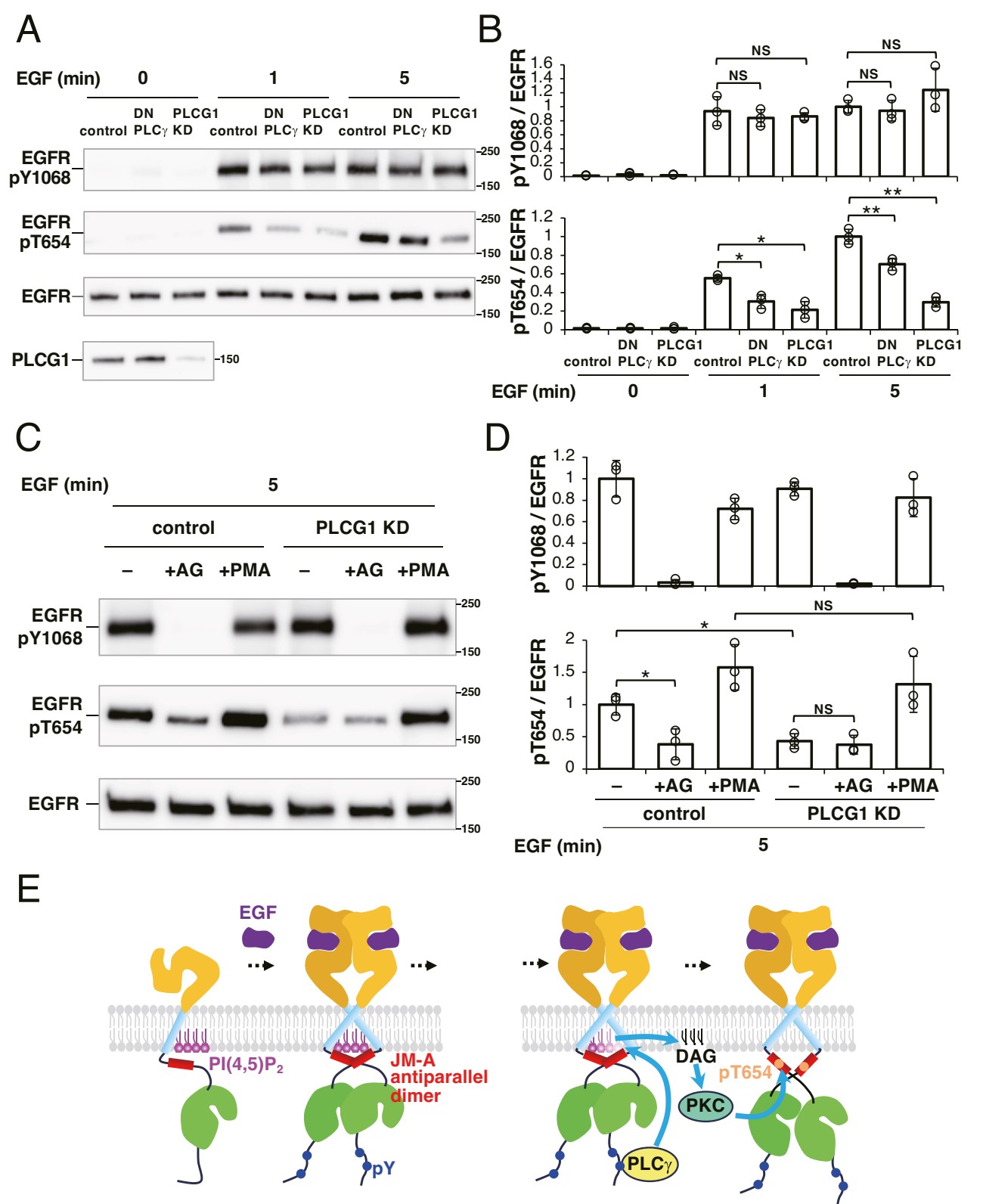

**Figure 7.** Thr654 of epidermal growth factor receptor (EGFR) is phosphorylated in a phospholipase Cγ (PLCγ)-dependent manner. (**A**) Effect of PLCγ on phosphorylation of EGFR. After EGFR-knockout (KO) cells were transfected with EGFR–Halo, EGFR–Halo and DN PLCγ, or EGFR–Halo and PLCG1 siRNA, the cells were incubated in serum-free medium overnight and stimulated with 10 nM EGF for 1 min or 5 min. DN PLCγ was transiently expressed in EGFR-KO cells expressing EGFR-Halo. Phosphorylated-Tyr1068 EGFR, phosphorylated-Thr654 EGFR, and total EGFR were detected with anti-EGFR

*Figure 7 continued on next page*

*Figure 7 continued*

pY1068, anti-EGFR pT654, and anti-EGFR antibodies, respectively. (**B**) Ratios of phosphorylated-Tyr1068 EGFR/total EGFR and phosphorylated-Thr654 EGFR/total EGFR. The ratios were normalized to the mean value of lane #7 in (**A**). Data are means ± SD of three experiments. (**C**) Effects of AG1478 and 4β-phorbol 12-myristate 13-acetate (PMA) on phosphorylation of EGFR. After EGFR-KO cells were transfected with EGFR–Halo or EGFR–Halo and PLCG1 siRNA, the cells were incubated in a serum-free medium overnight and stimulated with 10 nM EGF for 5 min in the presence or absence of 100 nM PMA. For inhibition of autophosphorylation of EGFR, cells were treated with 1 µM AG1478 for 30 min and stimulated with 10 nM EGF for 5 min. Phosphorylated-Tyr1068 EGFR, phosphorylated-Thr654 EGFR, and total EGFR were detected with anti-EGFR pY1068, anti-EGFR pT654, and anti-EGFR antibodies, respectively. (**D**) Ratios of phosphorylated-Tyr1068 EGFR/total EGFR and phosphorylated-Thr654 EGFR/total EGFR. The ratios were normalized to the mean value of lane #1 in (**C**). Data are means ± SD of three experiments. (**E**) Schematic model of the activation and signal transduction process for EGFR. Left, after EGF stimulation, PI(4,5)P$_2$ (magenta) interacts with the arginine residues in the JM-A region (red) of EGFR, stabilizing the JM-A antiparallel dimer, and inducing the formation of asymmetric dimers of the kinase region (green) of EGFR. This dimerization results in the phosphorylation of several tyrosine residues in the tail region of EGFR. Right, Phosphorylated-Tyr1068 EGFR recruits PLCγ (yellow), degrading PI(4,5)P$_2$ and producing DAG. DAG activates protein kinase C (cyan), which phosphorylates Thr654 of EGFR, deactivating EGFR. *p<0.05, **p<0.01, NS (not significant) on Welch's t-test.

The online version of this article includes the following source data for figure 7:

**Source data 1.** PDF file for western blotting analysis displayed in *Figure 7A and C*.

**Source data 2.** Original files for western blotting analysis are displayed in *Figure 7A and C*.

higher-order domain structure decreases after EGF stimulation. Single-molecule imaging confirmed these results, detecting the transient colocalization of EGFR and PI(4,5)P$_2$. The expression of a PI(4,5)P$_2$ phosphatase reduced the dimerization and autophosphorylation of EGFR after EGF stimulation. These findings are consistent with previous in vitro and in silico observations that PI(4,5)P$_2$ interacts strongly with the arginine residues in the JM-A region of EGFR, stabilizing the active conformation of EGFR (*Abd Halim et al., 2015*; *Matsushita et al., 2013*). Together with these observations, our biochemical analyses suggest that the local distribution of PI(4,5)P$_2$ around EGFR before EGF stimulation not only acts as the substrate of EGFR effectors, including PI3K and PLCγ, after EGF stimulation, but also plays a crucial role in enhancing the dimerization and kinase activation of EGFR immediately after EGF stimulation (*Figure 7E*, left).

We found that PI(4,5)P$_2$ is also involved in a specific pathway of EGFR deactivation after EGF stimulation. Tyr-phosphorylated EGFR recruits PLCγ, degrading PI(4,5)P$_2$ and producing DAG. DAG activates protein kinase C, which phosphorylates Thr654 of EGFR, deactivating EGFR (*Figure 7E*, right). We have reported previously that Thr654 phosphorylation causes the JM-A dimer to dissociate in the presence of acidic lipids (*Maeda et al., 2018*; *Maeda et al., 2022*). This constitutes the negative feedback regulation of the EGFR kinase function. Consistent with this, in our experiment, the expression of DN PLCγ did not induce DAG production (*Figure 6B*) or Thr654 phosphorylation (*Figure 7A and B*), but increased the Tyr1068 phosphorylation of EGFR (*Figure 7A and B*). After the degradation of PI(4,5)P$_2$, EGFR is free from both positive and negative regulation by PI(4,5)P$_2$. We expect this state to facilitate the higher-order oligomerization of EGFR, which is important for signaling to cytoplasmic proteins (*Hiroshima et al., 2018*; *Maeda et al., 2022*).

We found that of PI3K and PLCγ, PLCγ is responsible for the disaggregation of the EGFR and PI(4,5)P$_2$ nanodomains after EGFR phosphorylation. The difference between the effects of PI3K and PLCγ on the PI(4,5)P$_2$ nanodomains can be explained as follows. PLCγ is activated by EGFR after EGF stimulation (*Carpenter and Ji, 1999*), whereas PI3K is activated by ERBB3, which is associated with EGFR stimulation (*Hellyer et al., 1998*; *Mattoon et al., 2004*). This is supported by our results, which show that both ERBB3 and EGFR are required for EGF-induced AKT phosphorylation (*Figure 6—figure supplement 1C, D*). Therefore, the EGFR–EGFR–PLCγ and EGFR–ERBB3–PI3K complexes exist separately in the plasma membrane after EGF stimulation, which is consistent with the results in *Figure 6—figure supplement 1E and F*. Upon EGF stimulation, EGFR preferentially phosphorylates EGFR rather than ERBB3 (*Okada et al., 2022*). This may lead to higher activation of the EGFR–EGFR–PLCγ complex than the EGFR–ERBB3–PI3K complex. The small aggregation of PI(4,5)P$_2$ nanodomains in wortmannin-treated cells (*Figure 6—figure supplement 1A*) may indicate newly synthesized (*An et al., 2022*) or partially degraded PI(4,5)P$_2$ nanodomains at the plasma membrane after EGF stimulation.

EGF stimulation does not induce a significant change in PI(4,5)P$_2$ levels in the plasma membrane (*Delos Santos et al., 2017*). Here, it is shown that EGF stimulation reduces the density of small

PI(4,5)P$_2$ nanodomains in the plasma membrane around the EGFR. Together, these results indicate that PI(4,5)P$_2$ is hydrolyzed by PLCγ only in the region surrounding EGFR. In contrast, synaptojanin expression reduces PI(4,5)P$_2$ levels by 45% (*Field et al., 2005*), leading to defects in cytokinesis and mislocalization of PI(4,5)P$_2$-binding proteins to the plasma membrane (*Abe et al., 2012*; *Abe et al., 2021*; *Field et al., 2005*). These results suggest that synaptojanin expression decreases the amount of PI(4,5)P$_2$ throughout the plasma membrane rather than in the restricted region around EGFR.

We have shown here that the degree of coaggregation between EGFR and PI(4,5)P$_2$ is higher than that between EGFR and PS before EGF stimulation. The preferential coaggregation of the EGFR nanodomains with PI(4,5)P$_2$ rather than PS is probably due to the following two factors. The first is the curvature of the membrane. Previous electron microscopy–bivariate analyses showed that the nanodomains of inactive HRAS were enriched with PI(4,5)P$_2$, whereas those of active KRAS4B were enriched with PS due to highly specific RAS–lipid interactions determined by the membrane curvature (*Liang et al., 2019*; *Zhou et al., 2014*; *Zhou et al., 2015*). Like inactive HRAS, EGFR may prefer the PI(4,5)P$_2$ nanodomain environment, in addition to interacting electrostatically with PI(4,5)P$_2$. The second factor is tetraspanins, which form nanodomains at the plasma membrane. Tetraspanins may interact with EGFR and regulate EGFR activities and dynamics within tetraspanin nanodomains (*Berditchevski and Odintsova, 2016*; *Odintsova et al., 2003*; *Sugiyama et al., 2023*). Tetraspanins also indirectly interact with PI(4,5)P$_2$ (*Halova and Draber, 2016*). Therefore, EGFR may be localized in PI(4,5)P$_2$-enriched nanodomains mediated by tetraspanins. Further studies are required to reveal the relationships between these molecules.

PI(4,5)P$_2$ nanodomains may suppress EGFR signaling in the resting state despite their stabilizing effect on the JM-A dimer of EGFR during its kinase activation. Several reports have suggested that membrane cholesterol prevents EGFR phosphorylation before EGF stimulation (*Chen and Resh, 2002*; *Ringerike et al., 2002*). A biochemical analysis suggested that EGFR is associated with the detergent-resistant membrane fraction, which is enriched with cholesterol and PI(4,5)P$_2$ (*Brown and London, 1998*; *Hope and Pike, 1996*; *Laux et al., 2000*). We previously showed with SMLM that PI(4,5)P$_2$ nanodomains appear in the inner leaflet just beneath the sphingomyelin/cholesterol nanodomains located in the outer leaflet of the plasma membrane (*Abe et al., 2012*; *Makino et al., 2017*). Together with the results of the present study, this suggests that cholesterol- and PI(4,5)P$_2$-rich membrane domains exist in the plasma membrane, form higher-order aggregates, and transiently colocalize with EGFR clusters before EGF stimulation. Such membrane structures may prevent EGFR from activation because cholesterol exerts an inhibitory effect on EGFR. However, after EGF stimulation, rapid EGFR activation is induced by the positive effect of PI(4,5)P$_2$ on the dimerization of EGFR kinase. Transient complex of EGFR dimers with PI(4,5)P$_2$ may be required to leave the cholesterol nanodomains to allow kinase activation. The transient mobilization of EGFR clusters immediately after EGF stimulation that we have observed previously (*Hiroshima et al., 2018*) may reflect this process.

SMT analysis revealed that EGFR monomers decreased and dimers increased in the immobile fraction of control CHO-K1 cells after EGF stimulation (*Figure 4A*). The changes in oligomer size were smaller in the slow- and fast-mobile fractions than in the immobile fraction. In addition, the fraction size of the immobile state was not reduced after EGF stimulation (*Figure 3B*). These results suggest that stable EGFR dimer/oligomers were mostly increased in the immobile fraction after EGF stimulation. In contrast, a decrease in the colocalization rate between EGFR and PI(4,5)P$_2$ after EGF stimulation was observed in the slow- and fast-mobile fractions but not in the immobile fraction (*Figure 3—figure supplement 1D*). The PI(4,5)P$_2$ probes employed in the present study detect PI(4,5)P$_2$ near EGFR but they might not bind to PI(4,5)P$_2$ associated with EGFR due to steric hindrance. It is plausible that concentrated PI(4,5)P$_2$ molecules help to stabilize EGFR dimer/oligomers in the immobile fraction. However, we cannot exclude the possibility that the dimer/oligomers in the slow- and fast-mobile fractions were also stabilized by PI(4,5)P$_2$, which was not detected by the PI(4,5)P$_2$ probes.

For SMT analysis, we used EGFR–GFP rather than antibody-mediated labeling, as in previous studies (*Mudumbi et al., 2024*; *Sugiyama et al., 2023*). Compared with antibody-mediated labeling in the extracellular region of EGFR (*Sugiyama et al., 2023*), a higher change in EGFR mobility upon EGF stimulation was observed. The characteristics of EGFR or its binding efficiency to EGF may differ between the labeling methods. *Needham et al., 2016* and *Mudumbi et al., 2024* reported an increase in the size of single EGFR clusters after EGF stimulation as an indication of clustering using superresolution microscopy (*Mudumbi et al., 2024*; *Needham et al., 2016*). Such faint increases

were difficult to observe in our SMLM imaging under dense EGFR expression (*Figure 1*); however, SMT analysis detected increases in the singe-spot fluorescence intensities (*Figure 4*; *Hiroshima et al., 2018*; *Yasui et al., 2018*), which is consistent with EGF-induced EGFR clustering.

In conclusion, $PI(4,5)P_2$ plays a crucial role as a molecular switch for EGFR functions. $PI(4,5)P_2$ is involved in the complicated remodeling of the boundary lipids and higher-order membrane structure around EGFR. To detect whole these processes directly, further observations are required using cholesterol and DAG probes under SMLM or electron microscopy. Alternatively, further biochemical analyses measuring the local lipid contents around EGFR should be performed.

## Materials and methods

### Cell culture and drug treatments

HeLa cells (#RCB0007) and CHO-K1 cells (#RCB0285) were obtained from the RIKEN BioResource Research Center (Tsukuba, Japan). HeLa cells were grown at 37 °C in Dulbecco's modified Eagle's medium (DMEM) (Nacalai Tesque Inc, Kyoto, Japan) supplemented with 10% fetal bovine serum (FBS). We confirmed that all HeLa cell lines used in the present study were authenticated by STR profiling and were negative for mycoplasma contamination. CHO-K1 cells were grown at 37 °C in Ham's F-12 medium (FUJIFILM Wako Pure Chemical Corporation, Osaka, Japan), supplemented with 10% FBS. We confirmed that the CHO-K1 cells were authenticated by several gene sequences and were negative for mycoplasma contamination. All EGF stimulations were performed at 25 °C after preincubation of the cells at 25 °C for at least 10 min.

### Plasmid construction

DEST40/rsKame was generated by introducing the V157L mutation into Dronpa in DEST40/Dronpa (*Abe et al., 2012*). The coding sequence for human EGFR was cloned into DEST40/rsKame to generate DEST40/EGFR–rsKame. The coding sequence for the human GRP1-PH region was obtained from HeLa cell cDNA by PCR amplification and cloned into DEST40/PAmCherry1 to generate DEST40/PAmCherry–GRP1-PH. The coding sequence for the z-region of human PLCγ (DN PLCγ) was cloned into DEST40 (*Abe et al., 2012*) to generate DEST40/DN PLCγ. To generate DEST40/PAmCherry–PLCδ–PH–IRES–DN PLCγ, fragments of IRES and DN PLCγ were introduced immediately after PLCδ–PH in DEST40/PAmCherry– PLCδ–PH (*Abe et al., 2012*). pHalo–evt–2–PH and pHalo–TubbyC were generated by replacing the fragment of PLCδ–PH in pHalo–PLCδ–PH (*Abe et al., 2021*), with evt–2–PH and TubbyC, respectively. EGFR–Halo was generated by replacing the GFP fragment in EGFR–GFP (*Hiroshima et al., 2018*) with Halo7. EGFR(3RN)–Halo was generated by introducing R645N/R646N/R647N mutations into EGFR in EGFR–Halo. EGFRvIII–Halo was generated by deleting 30–297 amino acids of EGFR in EGFR–Halo. The coding sequence for human PLCG1 was obtained from HeLa cell cDNA by PCR amplification. PLCG1–SNAP was generated by replacing the EGFR and halo fragments in EGFR–Halo with PLCG1 and SNAP, respectively. The plasmids generated in the present study were deposited in the RIKEN BRC Gene Bank.

### Generation of EGFR or ERBB3-KO cells with CRISPR/Cas9 gene editing

To construct the gene-editing plasmids, oligomers (5'- caccgCACAGTGGAGCGAATTCCTT-3' and 5'- aaacAAGGAATTCGCTCCACTGTGc-3' for EGFR, 5'- caccgTGATCCAGCAGAGAACCCAG-3' and 5'- aaacCTGGGTTCTCTGCTGGATCAc-3' for ERBB3) were synthesized, annealed, and cloned into the PX459 vector (#48139, Addgene) at the BbsI site, as previously described (*Ran et al., 2013*). HeLa cells were transfected with the resultant plasmid using Lipofectamine 3000 (Thermo Fisher Scientific, Waltham, MA). Single colonies were picked after selection with 0.3 µg/ml puromycin for three days. The genomic DNAs were extracted from the clones with the GenElute Mammalian Genomic DNA Miniprep Kit (Sigma-Aldrich, St. Louis, MO). The genome sequences were examined by PCR using the following primers: 5'-GATCGTGGACATGCTGCCTCCTGTGTCCATGACTGC-3' and 5'-CTTCCCCT GCAGTATCTTACACACAGCCGGC-3' for EGFR KO, or 5'-aggtggggaaggcatctagggcaaaggg-3' and 5'-cggaactcgggcggtaatgcaagtgatgg-3' for ERBB3 KO cells. Sequence analysis revealed that EGFR KO had a 1 bp deletion (C) next to the protospacer adjacent motif (PAM) sequence and that ERBB3 KO had a 1 bp deletion (C) next to the PAM sequence.

### Establishing cell lines stably expressing EGFR–rsKame

To establish cell lines stably expressing EGFR–rsKame, EGFR KO cells were transfected with DEST40/EGFR–rsKame using Lipofectamine 3000 (Thermo Fisher Scientific) and cultured in the presence of 1 mg/ml G418 (Nacalai Tesque Inc) for 14 days. Stable clones expressing EGFR–rsKame were selected using fluorescence microscopy.

### Western blotting analysis

Parental HeLa and EGFR-KO cells stably expressing the EGFR–rsKame fusion protein were incubated in DMEM supplemented with 10% FBS. EGFR-KO cells were transfected with EGFR–GFP and incubated in DMEM supplemented with 10% FBS for 24 hr. EGFR-KO and ERBB3-KO cells were transfected with EGFR–Halo and incubated in DMEM supplemented with 10% FBS for 24 hr. The cells were further incubated in serum-free medium overnight and stimulated with 20 nM EGF for 1 min. After the cells were lysed in ice-cold RIPA lysis buffer (Nacalai Tesque Inc), the lysates were cleared by centrifugation at 15,000×g for 10 min at 4 °C. Protein concentrations were determined using a Pierce BCA Protein Assay Kit (Thermo Fisher Scientific). Proteins were applied on 4–15% TGX precast gels (Bio-Rad Laboratories, Hercules, CA) so that the protein concentrations were the same. The gels were transferred onto PVDF membranes using the Trans-Blot Turbo Transfer System and blocked with 2% non-fat dry milk in TBS + 0.1% Tween (TBST). Total EGFR was detected using anti-EGFR (RRID:AB_631420; #sc-3, Santa Cruz Biotechnology; 1:500) as the primary antibody and horseradish peroxidase (HRP)-linked anti-rabbit IgG (RRID:AB_2099233; #7074, Cell Signaling Technology, Danvers, MA; 1:1000) as the secondary antibody. Phosphorylated EGFR (pY1068) was detected using anti-EGFR pY1068 (RRID:AB_2096270; #3777, Cell Signaling Technology; 1:1000) as the primary antibody and HRP-linked anti-rabbit IgG. Total EGFR was detected as described above. Phosphorylated EGFR (pY1173) was detected using anti-EGFR pY1173 (RRID:AB_653167; #sc-12351, Santa Cruz Biotechnology, Dallas, TX, USA; 1:500) as the primary antibody and HRP-linked anti-goat IgG (RRID:AB_628490; sc-2354, Santa Cruz Biotechnology; 1:1000) as a secondary antibody. Phosphorylated AKT (pS473) was detected using anti-AKT (S473) (RRID:AB_2315049; #4060, Cell Signaling Technology; 1:1000) as the primary antibody and HRP-linked anti-rabbit IgG. Phosphorylated ERK (Thr202/Tyr204) was detected using anti-phospho-p44/42 (RRID:AB_2315112; #4370, Cell Signaling Technology; 1:1000) as a primary antibody and HRP-linked anti-rabbit IgG. Total ERK was detected using anti-p44/42 (RRID:AB_390779; #4695, Cell Signaling Technology; 1:1000) as the primary antibody and HRP-linked anti-rabbit IgG. Immunoreactive proteins were detected with ECL Prime Western Blotting Detection Reagent (GE Healthcare) using an ImageQuant LAS 500 device (GE Healthcare).

### Analysis of the effects of PLCγ and PKC on EGFR phosphorylation

EGFR-KO cells were transfected with EGFR(WT)–Halo, GFP–SYNJ, EGFR(3RN)–Halo, DN PLCγ, or PLCG1 siRNA (MISSION siRNA, SASI_Hs02_00334210, Sigma-Aldrich) using the Neon Transfection System (Thermo Fisher Scientific) and incubated in DMEM supplemented with 10% FBS for 24 hr. The medium was replaced with DMEM supplemented with 0.2% bovine serum albumin (BSA), and the cells were further incubated for 24 hr and stimulated with EGF. For PMA treatment, the cells were stimulated with EGF in the presence of 100 nM PMA. To inhibit autophosphorylation of EGFR, cells were pretreated with 1 μM AG1478 for 30 min and stimulated with EGF. Cells were washed twice with cold PBS. After the cells were lysed as described above, the proteins were separated on 4–15% TGX precast gels. Total EGFR was detected as described above. After the intensity of total EGFR was measured, we applied the proteins to SDS-PAGE again, so that the intensity of total EGFR in each sample was the same. Phosphorylated EGFR (pT654) was detected using anti-EGFR pT654 (RRID:AB_1523529; #ab75986, Abcam, Cambridge, UK; 1:500) as the primary antibody and HRP-linked anti-rabbit IgG. PLCG1 was detected using anti-PLCγ1 (RRID:AB_10691383; #5690, Cell Signaling Technology; 1:500) as the primary antibody and HRP-linked anti-rabbit IgG. Phosphorylated EGFR (pY1068) and total EGFR were detected, as described above.

### Measurement of DAG

DN PLCγ was transiently expressed in a cell line stably expressing EGFR–rsKame. The cells were incubated in a serum-free medium overnight and treated with or without 20 nM EGF for 1 min. The

amount of DAG in the DN PLCγ-expressing and control cells was measured using a DAG assay kit (#MET-5028, Cell Biolabs Inc, San Diego, CA) according to the manufacturer's instructions.

## Inhibition of PI3K

A cell line stably expressing EGFR–rsKame was incubated in a serum-free medium overnight and treated with or without 10 μM wortmannin for 1 hr. The cells were stimulated with or without 20 nM EGF for 1 min. Phosphorylated AKT was detected by western blotting analysis as described above. For SMLM imaging, after EGF stimulation, cells were treated with paraformaldehyde and glutaraldehyde.

## Analysis of EGFR dimerization

After EGFR KO cells were transfected with EGFR(WT)–Halo or EGFR(3RN)–Halo, the cells were incubated in DMEM supplemented with 10% FBS for 24 hr. The medium was replaced with DMEM supplemented with 0.2% BSA, and the cells were further incubated for 24 hr. The cells were incubated with 0.2 or 20 nM EGF for 2 min and washed twice with cold PBS. After the cells were incubated with 3 mM BS3 (Thermo Fisher Scientific) in PBS at 4 °C for 1 hr, BS3 was quenched with 20 mM Tris-HCl (pH 8.0). The cells were then washed twice with cold phosphate-buffered saline (PBS). After the cells were lysed, protein concentrations were determined. Proteins were subjected to western blotting analysis so that the protein concentrations were the same.

## Cy5–EGF labeling

After EGFR-KO cells were transfected with EGFR(WT)–Halo, EGFR(3RN)–Halo, or EGFRvIII–Halo, cells were incubated in DMEM supplemented with 10% FBS for 24 hr. The medium was replaced with DMEM supplemented with 0.2% BSA, and cells were further incubated for 24 hr. Cells were stained with 10 nM Janelia Fluor 549 (JF549) HaloTag Ligand (Promega, Madison, WI) for 30 min and were washed with the medium at least three times. After the medium was replaced with DMEM/Ham's F-12 containing HEPES and 0.2% BSA, 10 ng/mL Cy5–EGF was added as described previously (*Sako et al., 2000*). Confocal images were obtained within 10 min after Cy5–EGF with a confocal microscope (FV 3000, Olympus, Tokyo, Japan) equipped with a 60×1.35 NA objective lens.

## SMT analysis

SMT analysis was performed as previously described (*Kuwashima et al., 2021*; *Yanagawa et al., 2018*; *Yanagawa and Sako, 2021*) with modifications. For SMT analysis of EGFR and PI(4,5)P$_2$, HeLa cells were transfected with EGFR–GFP and Halo-PLCδ–PH. After the cells were incubated in DMEM supplemented with 10% FBS for 24 h, the medium was replaced with DMEM supplemented with 0.2% BSA and the cells were further incubated for 24 hr. Cells were stained with 10 nM JF549 for 30 min and washed with DMEM/Ham's F-12 containing HEPES and 0.2% BSA, at least three times. Single-molecule imaging of the living cells was performed at 25 °C. Fluorescently labeled proteins were observed under a microscope (TiE, Nikon, Tokyo, Japan). The cells were illuminated with a 488 nm laser (OBIS 488, Coherent Inc, Santa Clara, CA, USA) and a 561 nm laser (OBIS 561, Coherent). Fluorescent images of 150–200 frames were recorded using an Auto Imaging System (Zido, Osaka, Japan; http://zido.co.jp/en/) with an exposure time of 30 ms for dual-color imaging. The SMT and Variational Bayesian-Hidden Markov Model clustering analyses were performed with the Auto Analysis System (Zido) based on a two-dimensional Gaussian fitting algorithm, as described previously (*Kuwashima et al., 2021*; *Yanagawa et al., 2018*; *Yanagawa and Sako, 2021*). All subsequent analyses (diffusion dynamics, intensity distribution, colocalization, and statistical analysis) were performed using an updated version of the smDynamicsAnalyzer, Igor Pro 9.0, and a WaveMetrix (Igor)-based home-made program. Refer to the reference for detailed instructions and curve fitting functions (*Yanagawa and Sako, 2021*). Colocalization analysis was performed with slight modifications to account for the differences in expression levels among cells. After measuring the density of the particles (i.e. particle number/cell area), we normalized the original colocalization rate (*Yanagawa and Sako, 2021*) to the densities. We normalized all data for the colocalization rate and presented them as relative colocalization rates.

For SMT analysis of EGFR under PI(4,5)P$_2$-interaction-depleted conditions, CHO-K1 cells were transfected with EGFR–Halo, EGFR–Halo, and DEST40/GFP–SYNJ (*Abe et al., 2021*), or EGFR(3RN)–Halo and incubated in Ham's F-12 supplemented with 10% FBS for 24 hr. After replacing the medium with

Ham's F-12 supplemented with 0.2% BSA, the cells were further incubated for 24 hr. To observe EGFR in HeLa cells, EGFR-KO HeLa cells were transfected with EGFR–Halo or EGFR–Halo and DEST40/ GFP–SYNJ and incubated in DMEM supplemented with 10% FBS for 24 hr. After the medium was replaced with DMEM supplemented with 0.2% BSA, cells were further incubated for 24 hr. Cells were stained with 1 nM SaraFluor 650T (SF650) HaloTag Ligand (#A308-01, GORYO Chemical Inc, Sapporo, Japan) for 30 min and washed at least three times with DMEM/Ham's F-12 containing HEPES and 0.2% BSA. The fluorescently labeled proteins were observed using a 637 nm laser (OBIS 637, Coherent Inc). Expression of GFP–SYNJ was confirmed using a 488 nm laser. The fluorescence images were recorded as described above.

For SMT analysis of EGFR, PLCγ, and PI3K, EGFR-KO HeLa cells were transfected with EGFR–GFP, PLCG1–SNAP, and Halo–p85α and incubated in DMEM supplemented with 10% FBS for 24 hr. After the medium was replaced with DMEM supplemented with 0.2% BSA, cells were further incubated for 24 hr. Cells were stained with 10 nM SNAP-Cell TMR-Star ligand (#S9105S, New England Biolabs) and 1 nM SF650 HaloTag ligand for 30 min, and washed at least three times with DMEM/Ham's F-12 containing HEPES and 0.2% BSA. The fluorescently labeled proteins were observed under a micro-scope as described above with a 637 nm laser (OBIS 637, Coherent Inc). The fluorescently labeled proteins were observed with 488 nm laser, 561 nm laser, and 637 nm laser. The fluorescent images were recorded as described above.

## SMLM imaging

Cell lines stably expressing EGFR–rsKame were transfected with the plasmids using the Neon Trans-fection System (Thermo Fisher Scientific). After 1 day, the cells were incubated with 10 nM HMSiR HaloTag Ligand (#A201-01, GORYO Chemical, Inc) in DMEM supplemented with 0.2% BSA for 16 hr. The cells were then washed with fresh medium and incubated for 3 hr. The cells were fixed with 4% paraformaldehyde and 0.2% glutaraldehyde for 60 min and washed with PBS. To calibrate the drift, TetraSpeck Microspheres (#T7279; Thermo Fisher Scientific) were added before observation. Similar to the photoactivation of Dronpa, we photoactivated EGFR–rsKame on the basal plasma membrane with a 488 nm laser (OBIS 488, Coherent) for excitation and turning off the fluorescence, and attributed this to the spontaneous recovery of the stochastic turning-on of the fluorescence, instead of the illu-mination at 405 nm (*Mizuno et al., 2010*). PAmCherry was photoactivated with a 405 nm (COMPASS 405–25 CW, Coherent) and illuminated with a 532 nm laser (COMPASS 315 M-100, Coherent). HMSiR was illuminated with a 637 nm laser (OBIS 637, Coherent). EGFR–rsKame, PAmCherry, and HMSiR were observed under total internal reflection illumination using an inverted fluorescence microscope (TiE). 1000–2000 frames of fluorescent images were acquired using the Auto Imaging System (Zido, Japan) at a frame rate of 30 ms.

To stain phosphorylated EGFR, cells were fixed with 4% paraformaldehyde and 0.2% glutaralde-hyde for 60 min. Cells were permeabilized with 0.1% Triton X-100 in PBS and blocked with 2% BSA for 1 hr. The cells were then incubated with anti-EGFR pY1068 (RRID:AB_2096270; #3777, Cell Signaling Technology, 1:100) for 1 hr. The cells were then washed and incubated with HMSiR-coupled goat anti-rabbit IgG (#A204-01, GORYO Chemical, Inc, 1:250).

## SMLM analysis workflow

All subsequent analyses (image reconstitution, automated multi-distance spatial clustering analysis, G-SMAP analysis, and statistical analysis) were performed using an updated version of smDynam-icsAnalyzer. See the legend for *Figure 1—figure supplement 1* and *Yanagawa and Sako, 2021* for the details of the SMLM analysis workflow.

Localizations of single-fluorophore-labeled proteins (samples) and TetraSpeck Microspheres (beads) were determined by 2D Gaussian fitting algorism. Thermal drift and misalignment between color channels were corrected by an affine transformation using the localizations of the beads as indices (*Figure 1—figure supplement 1B*). The positional accuracy of the immobile fluorescence beads was estimated to be 6–8 nm (full width at half maximum, FWHM) (*Figure 1—figure supplements 1B and 2D*). The positional accuracy of the single molecules was typically 20–30 nm. The accuracy of the superimposition of two images after the affine transformation was 10–14 nm (*Figure 1—figure supplements 1B and 2E*).

SMLM images were reconstituted by convolving 2D-Gaussian functions for the single-molecule images (*Figure 1—figure supplement 1C*). To avoid the multi-counting of the same molecule, the positions of a single-molecule were averaged and plotted at a single location. The image resolution of SMLM was estimated with a method based on Fourier ring correlation (FRC) (*Nieuwenhuizen et al., 2013*). Resolution values for Dronpa(V157L), PAmCherry1, and HMSiR were 30.1±2.9 nm, 26.8±1.2 nm, and 28.2±1.0 nm, respectively (means ± SEM, n=5).

We developed and used a multi-distance spatial cluster analysis program to estimate the size and distribution of nano-domains automatically (*Figure 1—figure supplement 1D–F*), where the local point density around each point was compared with that for the random distribution based on the Ripley's K-function, K(r), and its variants for cluster analysis (*Kiskowski et al., 2009*) and co-cluster analysis (*Lagache et al., 2018*). We also generated a G-function spatial map (G-SMAP) to visualize single clusters (*Figure 1—figure supplement 1F*). The size of each cluster, the number of points within each cluster, and the point density in a cluster were calculated in the G-SMAPs.

## Statistical analysis

Statistical analyses were performed using Igor Pro software (WaveMetrics, Inc, Lake Oswego, OR) or Microsoft Excel (Microsoft, Redmond, WA). The experiments were repeated at least three times, as described in the figures and corresponding figure legends. The results are expressed as the means ± SEM. Statistical significance was determined using Tukey's multiple comparison test or Welch's t-test between the groups. Statistical significance was set at $p<0.05$, and they were classified into the following four categories: *$p<0.05$, **$p<0.01$, ***$<0.001$, and NS (not significant, $p\geq0.05$).

## Acknowledgements

This research was supported by Grants-in-Aid from the Ministry of Education, Culture, Sports, Science, and Technology of Japan (22K06609 to MA, 19H05647 to YS); Japan Science and Technology Agency (JST), PRESTO, JPMJPR20EF (MY); and the Glycolipidologue Program of RIKEN (to YS and TK). We thank Dr. Ryo Maeda for the technical advice for biological experiments. We also thank Maiko Minatohara for biochemical experiments. We thank the RIKEN BioResource Research Center for the STR profiling and mycoplasma tests. We thank the Support Unit for Bio-Material Analysis, RRD, and RIKEN for DNA sequence and cell sorting.

## Additional information

### Funding

| Funder | Grant reference number | Author |
| --- | --- | --- |
| Ministry of Education, Culture, Sports, Science and Technology | 22K06609 | Mitsuhiro Abe |
| Ministry of Education, Culture, Sports, Science and Technology | 19H05647 | Yasushi Sako |
| Japan Science and Technology Agency | 10.52926/JPMJPR20EF | Masataka Yanagawa |
| RIKEN | Glycolipidologue Program | Toshihide Kobayashi Yasushi Sako |

The funders had no role in study design, data collection and interpretation, or the decision to submit the work for publication.

### Author contributions

Mitsuhiro Abe, Funding acquisition, Investigation, Visualization, Writing - original draft; Masataka Yanagawa, Data curation, Software, Funding acquisition; Michio Hiroshima, Methodology; Toshihide Kobayashi, Resources, Funding acquisition; Yasushi Sako, Conceptualization, Supervision, Funding acquisition, Project administration, Writing - review and editing

## Author ORCIDs
Mitsuhiro Abe  https://orcid.org/0000-0002-6244-1771
Masataka Yanagawa  https://orcid.org/0000-0001-7832-0918
Yasushi Sako  https://orcid.org/0000-0002-5707-5455

## Decision letter and Author response
Decision letter https://doi.org/10.7554/eLife.101652.sa1
Author response https://doi.org/10.7554/eLife.101652.sa2

---

## Additional files

### Supplementary files
• MDAR checklist

### Data availability
All data generated or analysed during this study are included in the manuscript and supporting files; source data files have been provided for *Figure 1*, *Figure 1—figure supplement 2*, *Figure 2*, *Figure 3—figure supplement 1*, *Figure 4—figure supplement 2*, *Figure 5*, *Figure 6*, *Figure 6— figure supplement 1*, and *Figure 7*.

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
