## [Editor Report]

This paper reports a fundamental study explaining how signalling through a cell-surface receptor can lead to different cellular outputs through changing interactions with membrane lipids. The evidence supporting the conclusions is compelling, with state-of-the-art microscopy, including multicolour super-resolution microscopy and single-molecule imaging, coupled with carefully controlled experiments. This work will be of broad interest to cell biologists, particularly those with interests in cell signalling and lipid biology, in addition to researchers using imaging to study cellular phenomena.

---

## [Decision Letter]

[Editors' note: this paper was reviewed by Review Commons.]

---

## [Author Response]

General Statements

We are grateful to the reviewers for their critical comments and valuable suggestions, which have helped improve our manuscript. We have provided point-to-point answers to the comments and added detailed analyses to the revised manuscript.

Point-by-point description of the revisionsReviewer #1 (Evidence, reproducibility and clarity (Required)):The epidermal growth factor receptor (EGFR) is a central regulator of cell function, with important roles in development as well as tissue homeostasis in adults. The upregulation of EGFR expression or activity drives tumor progression in several types of cancer. This study examines the regulation of EGFR activity by compartmentalization of EGFR into unique plasma membrane nanodomains demarked by enrichment of phosphatidylinositol-4,5-bisphosphate (PIP2) and phosphatidylserine (PS). To do so, EGFR was labeled via C-terminal fusion with rsKame and labeling of PIP2 with PLCdelta-PH fused with PAmCherry or PS with evectin-2-PH-HaloTag. This was examined following fixation, using SMLM and statistical analysis using Ripley's univariate H-function (to determine the size and distribution of each nanocluster) and Ripley's bivariate H-function (to determine the distribution of different nanoclusters relative to one another). This revealed that the overlap of EGFR and PS did not change following stimulation of EGF, but this stimulation did reduce the overlap of EGFR and PIP2.These experiments in fixed cells were complemented with live cell single molecule imaging studies, tracking EGFR and PIP2. EGF stimulation increased the immobile fraction of EGFR, and decreased the overlap of EGFR and PIP2. Experiments with overexpression of the inositol 5-phosphatase synaptojanin and expression of mutants of EGFR (3RN) with a disrupted putative PIP2 binding site were used to assess the function of PIP2 in EGFR regulation. Expression of synaptojanin and mutation of EGFR (3RN) had similar results in that the confinement, dimerization, and c-terminal pY1068 of EGFR following EGF stimulation was altered. Perturbation of PIP2 availability by expression of a dominant interfering PLCgamma led to a reduction of the EGF-stimulated pT654 on EGFR, known to negatively regulate EGFR activation. From this emerges a model where clustering of EGFR with PIP2 nanodomains at the cell surface is required to support initial EGFR activation, and then the hydrolysis of PIP2 to generate DAG is required for eventual deactivation of EGFR.The manuscript is by Abe et al. is well-written, presenting a compelling investigation into the spatiotemporal dynamics of PIP2 and its regulatory role in EGFR activity in living cells. The use of super-resolution single-molecule microscopy to visualize the interactions between EGFR and PIP2 nanodomains and single particle tracking methodologies are complimentary, yielding important insights and underscoring the manuscript's contribution to advancing our understanding in this area. The experimental workflow is sophisticated and novel and is strengthened by the careful consideration of critical controls throughout the manuscript. For example, studying the expected Grb2 localization with EGFR, showing the expected gain in spatial correlation of EGFR and Grb2 upon EGF stimulation strengthens the use of this workflow to study interaction of EGFR and other nanoclusters. The results not only enhance our understanding of the intricate mechanisms governing EGFR regulation by lipids but also highlight the importance of PIP2 in the modulation of EGFR dimerization and autophosphorylation.While the experiments conducted in the study are largely well done and of high quality, there are several outstanding issues that must be addressed before considering publication. It is essential that the authors provide further clarification on certain aspects of their methodology for replicability and transparency. Additionally, a more detailed discussion on the implications of their findings within the broader context of cell signaling and potential impacts on related research areas would enhance the manuscript's significance. This could also require some additional experiments to align the observations made in this study with that of previous studies, in particular as it relates to PIP2 dynamics and clustering. Addressing these concerns will not only strengthen the conclusions drawn but also provide the scientific community with a more comprehensive understanding of the complex interplay between putative PIP2 nanodomains and EGFR activity. The resolution of these issues is crucial for the manuscript to fully meet the publication standards of contributing novel and impactful insights to the field.Major comments1. In Figure 3B, the dramatic (~20%) change in EGFR mobility upon EGF stimulation (i.e. from fast-mobile to confined) implies EGFR-EGF binding in excess of what is typically reported in the literature. How do the authors reconcile this and are there features of their cell model/analysis pipeline that are artificially contributing to this observation?

We tagged the cytoplasmic tail of EGFR with GFP (EGFR–GFP) (p. 11, l.198–201) instead of using antibody-mediated labeling, which has been used in previous studies. Compared with antibody-mediated labeling in the extracellular region of EGFR, a greater change in EGFR mobility upon EGF stimulation was observed, which is consistent with previous studies (Hiroshima et al., 2018; Yasui et al., 2018) (p. 11, l.209–211). The differences in labeling may cause differences in the characteristics of EGFR or binding efficiency to EGF. We have mentioned this possibility in the Discussion section (p. 25, l.501–505).

2. The detection of PIP2 nanodomains in the plasma membrane is somewhat controversial, especially using the PH domain of PLCd to detect PIP2 or using similar strategies. The recent study by Pacheco et al. (JCB 2023, PMID: 36416724) uses a variety of measurements of fluorescent labeling of PIP2 by protein-based biosensors (similar to this study) and concludes that PIP2 is free diffusing in the plasma membrane, which would be inconsistent with PIP2 nanodomains. Pacheco et al. propose that while engagement of PIP2 to effectors via the inositol headgroup may serve to immobilize this lipid, allowing clustering, the use of relatively large protein domains as fluorescent ligands that bind to the PIP2 headgroup to track PIP2, as performed here, displaces any intrinsic clustering mechanism, leading to free diffusion of PIP2. How can the clustering observed here for PIP2 be reconciled? Is it possible that additional, non lipid-based interactions function alongside PH domain-PIP2 interactions as a form of coincidence detection? It would be quite helpful to support the data shown in this manuscript with a different PIP2 binding domain, such as the Tubby domain used by Pacheco et al. It would not be necessary to repeat all experiments with such a complementary probe, but some key experiments that assess the apparent clustering of PIP2 would be important to consider repeating with this complementary PIP2 probe.

As suggested by the reviewer, we performed SMLM and SMT analyses using TubbyC. No significant differences were observed between PLCd–PH and TubbyC probes. These results are shown in Figure. 2C (p. 9, l.167–172), and Figure 3—figure supplement 1E, 1F (p. 12, l.226–229).

3. It is unclear if and how stimulation with EGF or overexpression of synaptojanin modulates PIP2 in the plasma membrane. Some studies found that EGF stimulation does not change PIP2 levels in the PM, including Delos Santos et al. (Mol Biol Cell. 2017, PMID: 28814502). Others found that the regulation of PIP2 levels in the plasma membrane is tightly controlled and the total levels of PIP2 can resist alterations of PIP2 by changes in lipid enzymes (Wills et al. JCS 2023, PMID: 37534432). Hence, it is not clear if the stimulation with EGF or the overexpression of synaptojanin changes plasma membrane PIP2 levels, or may only alter the nanoscale dynamnics of this lipid. If the effects of synaptojanin may be restricted to alterations of the nanoscale organization of PIP2 in the membrane, it would be important to consider that synaptojanin is strongly localized to clathrin-coated pits in the plasma membrane (e.g. Perera et al. PNAS 2007. PMID: 17158794), and that EGFR only exhibits strong recruitment to clathrin-coated pits following EGF stimulation, which would suggest that the non-ligand-bound EGFR is distant to synaptojanin-containing structures. Some consideration of the possibility of broad action of PIP2 depletion vs nanoscale localized effects by these treatments should be considered when interpreting the results of this study.

Here, we show that EGF stimulation decreases the degree of PI(4,5)P_2_ nanodomain clustering, mainly by reducing the density of small nanodomains. As noted by the reviewer, EGF stimulation does not induce a significant change in PI(4,5)P_2_ levels in the plasma membrane (Delos Santos et al., 2017). These results suggest that PI(4,5)P_2_ may be hydrolyzed by PLCγ only in the region around EGFR. In contrast, synaptojanin expression reduced PI(4,5)P_2_ levels by 45% (Field et al., 2005), leading to defects in cytokinesis (Abe et al., 2012; Field et al., 2005). We previously found that synaptojanin expression diminishes the localization of PI(4,5)P_2_-binding proteins to the plasma membrane (Abe et al., 2012; Abe et al., 2021). These results suggest that synaptojanin expression decreases the amount of PI(4,5)P_2_ throughout the plasma membrane rather than in the restricted region around EGFR. We have added this information to the Discussion section (p. 22, l.443–452).

Minor comments1. Please quantify the extent to which endogenous EGFR was knocked down.

We knocked out rather than knocked down the EGFR gene with CRISPR/Cas9 gene editing. Therefore, there was no intrinsic EGFR in the cells used (Figure 1—figure supplement 2A). The detailed methods is described in the Methods section (p. 28, l.546–564).

2. Figure 2C – please provide the entire field of view, including the area chosen for the zoomed in images.

As suggested by the reviewer, we have added the entire and enlarged images to the new Figure 2D (p. 9, l.173–174).

3. Regarding the time points chosen to measure EGFR area and others. Why were the 1 mins and 2 mins time points chosen to examine EGFR-PIP2 and EGFR-GRB2 interactions, respectively? Is there evidence that these interactions peak at these time points? Alternatively, please provide evidence of their interactions at earlier time points (e.g. 15-30 seconds for EGFR-PIP2 and 1 mins for EGFR-GRB2).

SMT analysis indicated that the colocalization rate of EGFR and PI(4,5)P_2_ significantly decreased after 0.5 min of EGF stimulation. For EGFR and GRB2, the rate was reduced after 0.5 min of EGF stimulation. These results suggest that the binding of EGF to EGFR and the subsequent reaction occurred within 0.5 min under our experimental conditions. We have added the relevant data to Figure 3—figure supplement 2 (p. 12, l.229–234). In addition, we have provided the bivariate H(r) values of EGFR and PI(4,5)P_2_ at 0.5 min after EGF stimulation, as obtained with SMLM analysis (Figure 2B, green) (p. 9, l.166–167).

4. Please demonstrate with immunoblot the extent to which EGFR-EGFP construct can be stimulated by EGF (EGFR Y1068) vs. control cells.

As suggested by the reviewer, we have added the results to the new Figure 3—figure supplement 1A. Although we reduced the expression level of EGFR–GFP for SMT, it was phosphorylated to the same level after EGF stimulation as the intrinsic EGFR in the parental cells (p. 11, l.201–203).

5. Figure 3—figure supplement 1B (right) – how do the authors explain the apparent decrease in diffusion coefficient for the fast mobile fraction of EGFR. Presumably, these receptors are not engaged with ligand, so what is causing the decrease in diffusion coefficient?

It is not necessary to assume that the EGFR in the fast mobile fraction was EGF-free. In the Variational Bayesian-Hidden Markov Model analysis, all time points in the single trajectories were assigned to one of the three states, and we frequently observed state transitions within a single trajectory (Hiroshima et al. 2018). These transitions suggest that the difference in the mobility states is not caused solely by ligand binding, as differences in the membrane environment might also influence the mobility. Therefore, ligand-free and ligand-bound molecules have different diffusion coefficients in the same fast-mobile fraction.

6. Figure 6A – it is unclear which method was used to probe pAKT (S473) inhibition by wortmannin. Please specify.

The cells were incubated overnight in serum-free medium, treated with 10 µM wortmannin for 1 h, and stimulated with 20 nM EGF. pAKT was detected using anti-AKT (S473) (#4060, Cell Signaling Technology) as the primary antibody. This is described in the Methods section (p. 30, l.591–592; p. 31, l.623–627).

7. Figure 6E (middle) – The authors explain that wortmannin treatment causes PIP2 dispersal. This interpretation would be strengthened with a quantification, as another interpretation of the representative image is that wortmannin appears to reduce the abundance of PIP2. This discrepancy requires explanation. There also appears to be an increase in pEGFR Y1068 relative to control.

The extent of PI(4,5)P_2_ distribution after wortmannin treatment was not a focus of this study; consequently, we deleted the text “wortmannin treatment causes PI(4,5)P_2_ dispersal.” In addition, we replaced the images and changed the pseudocolor to avoid the incorrect impression that the amount of pEGFR was increased in wortmannin-treated cells (Figure 6E).

8. Please provide uncropped immunoblot images without contrast adjustment. Some immunoblots appear to have lane-to-lane differences in background.

For all immunoblot images in this study, we did not adjust the contrast or brightness of the original images. The figure source data files show the images before they were trimmed.

9. Given the conclusion on line 421 re: PI3K localization, can you provide data to support that this is the case (i.e. that PI3K acts at distinct sites on the membrane away from the specific PIP2-EGFR nanodomains)? This should be possible given the methods described in the manuscript.

We want to say that PLCγ hydrolyzes PI(4,5)P_2_ around EGFR, whereas PI3K phosphorylates PI(4,5)P_2_ around the heterodimer of ERBB3 and EGFR in the plasma membrane. We have added the results of the SMT analysis, which indicated that the colocalization rate of PLCγ–PI3K was lower than that of EGFR–PLCγ or EGFR–PI3K (Figure 6—figure supplement 1E, 1F) (p. 18, l.366–369). We have also added this information to the Discussion section (p. 22, l.428–442).

10. Likewise, showing whether this specific pool of PIP2-EGFR nanodomains are within or away from the well-characterized EGFR-tetraspanin nanodomains would add value to the interpretation of the results. However, this reviewer notes that this would add significant experiments to the study and this could be considered in future studies.

We were also interested in the local content of PI(4,5)P_2_ in EGFR–tetraspanin nanodomains. In the revised Discussion section we state that further studies will reveal the relationship between these molecules (p. 24, l.465–468).

11. Please explicitly state whether statistical considerations were made for multiple comparisons in the methods.

As the reviewer suggested, we have added the description in the Methods section (p.38, l.753-754).

Reviewer #1 (Significance (Required)):General strengths:- This study examines an important question in the cell biology of a key regulator of cell physiology, EGFR. While the mobility and nanodomain clustering of EGFR has emerged as critical for the regulation of this receptor's ligand binding, dimerization, and downstream signaling output, there remains much to be understood about the nanoscale organization of EGFR relative to other signaling lipids and proteins in the plasma membrane. This study examines a novel link between nanodomains demarked by PIP2 and EGFR mobility and signaling.- This study makes use of sophisticated multiplexed labeling of EGFR and lipids such as PIP2 and PS, along with super-resolution microscopy and single molecule imaging coupled to cutting-edge image quantification.- Controls are generally well-considered and appropriate to support and validate the experimental workflows that areGeneral limitations:- The labeling of PIP2 with fluorescently-labelled protein domains that recognize this lipid has some limitations, as described above in the comments.Advance: This study fills a knowledge gap in how the central regulator of cell physiology, EGFR, is organized at the cell surface. It is well appreciated that EGFR exhibits confinement at the plasma membrane and that this receptor exhibits nanoscale clustering that regulates receptor function. However, the nature of the nanoscale clusters in which EGFR is detected in the ligand-bound and non-ligand-bound states, and how this defines receptor output is only beginning to be resolved. This study examined how clustering of the lipid PIP2 in the plasma membrane relates to EGFR clusters, and how this may functionally impact EGFR signaling. This fills a knowledge gap of the molecules within the plasma membrane that impact receptor nanoscale clustering and function. This study also advances how mechanisms that impact EGFR nanoscale organization also in turn affect signaling output, which provides compelling evidence for the significance of EGFR-PIP2 interactions (especially with the EGFR mutant that is predicted to have reduced PIP2 interactions).Audience: This study will be of significant interest to fundamental cell biology researchers in general, and in particular those interested in cell signaling and lipid cell biology.Reviewer expertise: This reviewer has expertise in cell biology of receptor signaling, phosphoinositides, single-particle tracking, and plasma membrane nanodomains.Reviewer #2 (Evidence, reproducibility and clarity (Required)):The work presented in "Bilateral regulation of EGFR activity and local π dynamics observed with superresolution microscopy" by Abe et al. studies the role of PI(4,5)P2 in EGFR signaling. This is an important question since the interplay of lipids and membrane receptors is known to be important for signaling, but the underlying mechanisms are not fully understood. The authors use multicolor superresolution and single molecule tracking coupled with biochemical approaches to understand how EGFR and PIP2 interplay on the plasma membrane. This work focuses on the biophysical mechanism of EGFR signaling and will be relevant to journals in the areas of biophysics, cell biology, cell signaling and microscopy. Overall, this an important study that identifies PIP2 as playing a functional role in EGFR signaling. However, there are some caveats to the experimental conditions that need to be discussed.Minor comments:From the original paper (Rosenbloom et al), it seems that rsKame still requires photoactivation at 405 nm. Was the done here for superresolution imaging? It is not listed in the methods for rsKame.

Similar to the photoactivation of Dronpa (Mizuno et al., 2010), we photoactivated rsKame with a 488 nm laser for excitation and turning off the fluorescence, and we attributed this to the spontaneous recovery of the stochastic turning on of the fluorescence, instead of the illumination at 405 nm. We have added this information to the Methods section (p. 35, l.705-709).

The stimulation conditions vary throughout, in both EGF concentration and time (1-5 min), possible differences due to various stimulation conditions should be discussed. Furthermore, superresolution samples were fixed after 1 min of EGF stimulation. The lack of EGFR reorganization may be due to the time required for EGF to diffuse to the adherent cell surface. Other superresolution imaging has demonstrated that EGFR forms oligomers on the cell after EGF stimulation (e.g., Mudumbi et al., Cell Reports 2024; Needham et al. Nat Comm 2016). Comment if your results are consistent or not with these other works.

For SMT and SMLM analyses throughout this study, the cells were treated with 20 nM EGF, which is sufficiently above the dissociation constant of 2–6 nM (Sugiyama et al., 2023). In the revised manuscript, we examined the time course of colocalization between EGFR and PI(4,5)P_2_ or GRB2 (Figure 3—figure supplement 2) (p. 12, l.229–234). The colocalization rate of EGFR and PI(4,5)P_2_ decreased significantly by 0.5 min after EGF stimulation and remained low for at least 5 min (Figure 3—figure supplement 2A). Under the same experimental conditions, the colocalization rate of EGFR and GRB2 increased by 0.5 min after EGF stimulation and remained high for at least 5 min (Figure 3—figure supplement 2B). These results suggest that the binding of EGF to EGFR and the subsequent reaction were almost in a steady state, at least during 0.5–5 min of EGF stimulation.

As noted by the reviewer, Mudumbi et al. showed that the cluster size of EGFR increased after EGF stimulation, which may be different from our results (Figure 1B). However, their methods and aims were different from those of the SMLM analysis in this study. They used very sparse conditions for single-cluster analysis, whereas in SMLM we used dense conditions to detect the coaggregation of nanodomains, which may contain multiple clusters. We also used sparse conditions for single-molecule imaging (Figure 4) and observed dimer/oligomer formation with an increase in the fluorescence signal in the single EGF spots. We observed a similar increase in the single-spot fluorescence signal in our previous studies. Our results are at least qualitatively consistent with those reported previously (Mudumbi et al., 2024; Needham et al., 2016). We have added this information to the Discussion section (p. 25, l.506–512).

Single molecule tracking was performed at room temperature. At what temperature were the superresolution and western blot samples stimulated? Fluidity and organization of the plasma membrane is altered by temperature. Possible caveats should be discussed. If biochemistry was performed at 37 C, then EGFR signaling cannot be correlated between samples dues to faster activation at physiological temperature.

Cells were stimulated with EGF at 25°C in all experiments. We have added this information to the Methods section (p. 27, l.525–526).

Many experiments are performed using transient transfection, with no control for or quantification of expression level. The frequency of EGFR:domain overlap and colocalization during SMT could be dependent on the relative expression levels of proteins/lipid markers. Was this accounted for?

As noted by the reviewer, the expression levels of EGFR and probes differ among cells. To consider the differences in expression levels among cells, we measured the density of particles (particle number/cell area). We then normalized the original colocalization rates, which were calculated using our custom-made software (Yanagawa and Sako, 2021), to the densities of both the EGFR and the probes. We normalized all data for the colocalization rates and presented them as relative colocalization rates (p. 12, l. 220–223; p. 34, l. 669–673)

Please describe why some experiments performed with HeLa EGFR knockout cells and other with CHO cells?

As noted by the reviewer, we used CHO-K1 cells in the experiment shown in Figure 4, whereas HeLa cells were used in other experiments. We performed a similar experiment using HeLa cells and we obtained similar results. The results for HeLa cells are presented in Figure 4—figure supplement 1A (p. 13, l.256–257).

The author state that "…dimerized EGFR was mainly found in the immobile fraction…" How did they determine this? This interpretation of the SMT data is important for suggesting that PIP2 EGFR stabilization (Figure 4), and should therefore be explain/justified.

As shown in Figure 4A, SMT analysis revealed that EGFR monomers decreased and dimers increased in the immobile fraction of control CHO-K1 cells after EGF stimulation. The changes in oligomer size were smaller in the slow- and fast-mobile fractions than in the immobile fraction. In addition, the fraction size of the immobile state was not reduced after EGF stimulation (Figure 3B). These results suggest that stable EGFR dimer/oligomers were mostly increased in the immobile fraction after EGF stimulation. In contrast, the colocalization rate in the slow-mobile and fast-mobile fractions decreased after EGF stimulation, whereas the rate in the immobile fraction did not change significantly (Figure 3—figure supplement 1D). The PI(4,5)P_2_ probes employed in this study can detect PI(4,5)P_2_ near EGFR but they might not bind to PI(4,5)P_2_ associated with EGFR due to steric hindrance. It is plausible that concentrated PI(4,5)P_2_ molecules help to stabilize EGFR dimer/oligomers in the immobile fraction. However, we cannot exclude the possibility that the dimer/oligomers in the slow- and fast-mobile fractions were also stabilized by PI(4,5)P_2_, which was not detected by the PI(4,5)P_2_ probes. We have added this information to the Discussion section (p.25, l.487-500).

In Figure 3, it is shown that the immobile fraction of EGFR increases with EGF, while PIP2 diffusion is unchanged. If PIP2 interacts with EGFR and stabilizes dimers, would you expect to also see an increase in the PIP2 immobile fraction?

Our results suggest that the interaction between EGFR and PI(4,5)P_2_ is transient (Figure 3D) (p.11, l.218-220). Therefore, we did not observe an increase in the PI(4,5)P_2_ immobile fraction. We have added a movie to the revised manuscript. Video 1 shows the lateral colization of EGFR and PI(4,5)P_2_ before and after EGF stimulation. As mentioned above, despite the decrease in PI(4,5)P_2_ concentration after EGF stimulation, colocalization between EGFR and PI(4,5)P_2_ was maintained in the immobile fraction.

Reviewer #2 (Significance (Required)):This study address an important question since the interplay of lipids and membrane receptors is known to be important for signaling, but the underlying mechanisms are not fully understood. The authors are able to make a conceptual advance in our understanding of EGFR biology by using advanced imaging techniques that allow for quantification of protein distribution and dynamics on intact cells. The elegant application of superresolution and single molecule tracking are important strengths of this work, while the clever use of receptor mutant and phosphates reveals novel insights. This work focuses on the biophysical mechanism of EGFR signaling and will be relevant to journals in the areas of biophysics, cell biology, cell signaling and microscopy.

References

Abe, M., A. Makino, F. Hullin-Matsuda, K. Kamijo, Y. Ohno-Iwashita, K. Hanada, H. Mizuno, A. Miyawaki, and T. Kobayashi. 2012. A role for sphingomyelin-rich lipid domains in the accumulation of phosphatidylinositol-4,5-bisphosphate to the cleavage furrow during cytokinesis. *Mol Cell Biol*. 32:1396-1407.

Abe, M., A. Makino, M. Murate, F. Hullin-Matsuda, M. Yanagawa, Y. Sako, and T. Kobayashi. 2021. PMP2/FABP8 induces PI(4,5)P(2)-dependent transbilayer reorganization of sphingomyelin in the plasma membrane. *Cell Rep*. 37:109935.

Delos Santos, R.C., S. Bautista, S. Lucarelli, L.N. Bone, R.M. Dayam, J. Abousawan, R.J. Botelho, and C.N. Antonescu. 2017. Selective regulation of clathrin-mediated epidermal growth factor receptor signaling and endocytosis by phospholipase C and calcium. *Mol Biol Cell*. 28:2802-2818.

Field, S.J., N. Madson, M.L. Kerr, K.A. Galbraith, C.E. Kennedy, M. Tahiliani, A. Wilkins, and L.C. Cantley. 2005. PtdIns(4,5)P2 functions at the cleavage furrow during cytokinesis. *Curr Biol*. 15:1407-1412.

Hiroshima, M., C.G. Pack, K. Kaizu, K. Takahashi, M. Ueda, and Y. Sako. 2018. Transient Acceleration of Epidermal Growth Factor Receptor Dynamics Produces Higher-Order Signaling Clusters. *J Mol Biol*. 430:1386-1401.

Mizuno, H., P. Dedecker, R. Ando, T. Fukano, J. Hofkens, and A. Miyawaki. 2010. Higher resolution in localization microscopy by slower switching of a photochromic protein. *Photochem Photobiol Sci*. 9:239-248.

Mudumbi, K.C., E.A. Burns, D.J. Schodt, Z.O. Petrova, A. Kiyatkin, L.W. Kim, E.M. Mangiacapre, I. Ortiz-Caraveo, H. Rivera Ortiz, C. Hu, K.D. Ashtekar, K.A. Lidke, D.S. Lidke, and M.A. Lemmon. 2024. Distinct interactions stabilize EGFR dimers and higher-order oligomers in cell membranes. *Cell Rep*. 43:113603.

Needham, S.R., S.K. Roberts, A. Arkhipov, V.P. Mysore, C.J. Tynan, L.C. Zanetti-Domingues, E.T. Kim, V. Losasso, D. Korovesis, M. Hirsch, D.J. Rolfe, D.T. Clarke, M.D. Winn, A. Lajevardipour, A.H.A. Clayton, L.J. Pike, M. Perani, P.J. Parker, Y. Shan, D.E. Shaw, and M.L. Martin-Fernandez. 2016. EGFR oligomerization organizes kinase-active dimers into competent signalling platforms. *Nature Communications*. 7:13307.

Sugiyama, M.G., A.I. Brown, J. Vega-Lugo, J.P. Borges, A.M. Scott, K. Jaqaman, G.D. Fairn, and C.N. Antonescu. 2023. Confinement of unliganded EGFR by tetraspanin nanodomains gates EGFR ligand binding and signaling. *Nat Commun*. 14:2681.

Yanagawa, M., and Y. Sako. 2021. Workflows of the Single-Molecule Imaging Analysis in Living Cells: Tutorial Guidance to the Measurement of the Drug Effects on a GPCR. *Methods Mol Biol*. 2274:391-441.

Yasui, M., M. Hiroshima, J. Kozuka, Y. Sako, and M. Ueda. 2018. Automated single-molecule imaging in living cells. *Nat Commun*. 9:3061.